# Partial Correlation Network Estimation by Semismooth Newton Methods

**Dongwon Kim**[*]
Department of Statistics
Seoul National University
dongwonida@snu.ac.kr

**Sungdong Lee**[*]
Department of Medicine
National University of Singapore
sdlee23@nus.edu.sg

**Joong-Ho Won**
Department of Statistics
Seoul National University
won.j@snu.ac.kr

## Abstract

We develop a scalable second-order algorithm for a recently proposed $\ell_1$-regularized pseudolikelihood-based partial correlation network estimation framework. While the latter method admits statistical guarantees and is inherently scalable compared to likelihood-based methods such as graphical lasso, the currently available implementations rely only on first-order information and require thousands of iterations to obtain reliable estimates even on high-performance supercomputers. In this paper, we further investigate the inherent scalability of the framework and propose locally and globally convergent semismooth Newton methods. Despite the nonsmoothness of the problem, these second-order algorithms converge at a locally quadratic rate, and require only a few tens of iterations in practice. Each iteration reduces to solving linear systems of small dimensions or linear complementary problems of smaller dimensions, making the computation also suitable for less powerful computing environments. Experiments on both simulated and real-world genomic datasets demonstrate the superior convergence behavior and computational efficiency of the proposed algorithm, which position our method as a promising tool for massive-scale network analysis sought for in, e.g., modern multi-omics research.

## 1 Introduction

A partial correlation network refers to a graph in which each node (vertex) corresponds to a variable and an edge between two nodes represents the partial corrrelation between the corresponding pair of variables. Partial correlation is defined as the correlation coefficient between the residuals resulting from the linear regression of each of the two variables with the rest of the variables, thereby measuring the degree of association between the two adjusting for the effect of the remaining variables. Learning partial correlation networks form data is particularly important in multi-omics studies of biological systems, where spurious edges due to shared regulatory factors should be avoided to detect direct biological interaction of causal implications [27, 24, 22].

Partial correlation network estimation is closely connected to covariance selection [6], which estimates nonzero entries of the inverse covariance or precision matrix of the variables. This is because a nonzero entry of the precision matrix implies a nonzero partial correlation between the corresponding variables. Moreover, if normality of the data distribution is assumed, then the zero entries of the precision matrix imply conditional independence. For this reason, when the number of variables is large and it is reasonable to assume that the number of edges is small, the graphical lasso [8, 34] that maximizes the Gaussian likelihood under $\ell_1$ penalty on the entries of the precision matrix or solve the following optimization problem

$$\min_{\boldsymbol{\Theta} \in \mathbb{S}^p} -\log \det \boldsymbol{\Theta} + \mathbf{tr}(\boldsymbol{S}\boldsymbol{\Theta}) + \lambda \|\boldsymbol{\Theta}_{-D}\|_1, \lambda > 0, \tag{1}$$

---

[*]Equal contribution

39th Conference on Neural Information Processing Systems (NeurIPS 2025).

is by far the most popular choice. Here, $\mathbb{S}^p$ denotes the space of $p$-dimensional symmetric matrices, and $\boldsymbol{S} = (1/n)\boldsymbol{X}^T\boldsymbol{X}$ is the sample covariance matrix of the data $\boldsymbol{X} = [x_1, \ldots, x_n]^\top \in \mathbb{R}^{n \times p}$. The $\boldsymbol{\Theta}_{-D}$ denotes a matrix that takes the off-diagonal elements of $\boldsymbol{\Theta}$ with zero diagonal elements. The $\ell_1$ norm $\|\cdot\|_1$ sums up the absolute values of the elements of its input matrix. The solution $\hat{\boldsymbol{\Theta}}$ to (1) estimates the target precision matrix $\boldsymbol{\Theta}^*$, such that $x_i \overset{iid}{\sim} N(0, (\boldsymbol{\Theta}^*)^{-1})$.

However, many methods that directly solve (1) suffer from computational bottleneck when $p$ reaches a few thousands, which is not desirable for analyzing modern massive-scaled data such as arising from multi-omics. The root cause is that the Karush-Kuhn-Tucker (KKT) optimality condition of (1)

$$-\boldsymbol{\Theta}^{-1} + \boldsymbol{S} + \lambda\boldsymbol{Z} = \boldsymbol{0}, \quad \boldsymbol{Z} \in \partial\|\boldsymbol{\Theta}_{-D}\|_1, \tag{2}$$

where $\partial\|\boldsymbol{\Theta}_{-D}\|_1$ denotes the subdifferential of the $\ell_1$ norm at $\boldsymbol{\Theta}_{-D}$, involves matrix inversion. Direct computation of the inverse $\boldsymbol{\Theta}^{-1}$ requires $O(p^3)$ arithmetic operations. Since most existing algorithms for solving (1) (e.g., [5, 8, 20, 13]) can be understood as solving (2) in an iterative fashion, such a costly computation essentially occurs every iteration. Worse, matrix inversion is difficult to parallelize or distribute, limiting scalability.

A recently proposed framework called ACCORD [19] overcomes this limitation by employing a computation-friendly loss function in a one-to-one transformation of the precision matrix variable $\boldsymbol{\Theta}$. It solves a convex optimization problem

$$\min_{\boldsymbol{\Omega} \in \mathbb{R}^{p \times p}} -\log\det\boldsymbol{\Omega}_D + (1/2)\operatorname{tr}(\boldsymbol{\Omega}^T\boldsymbol{\Omega}\boldsymbol{S}) + \lambda\|\boldsymbol{\Omega}_{-D}\|_1, \tag{3}$$

where $\boldsymbol{\Omega}_D$ denotes the diagonal matrix that takes the diagonal elements of $\boldsymbol{\Omega}$. Precision matrix $\boldsymbol{\Theta}$ and the optimization variable $\boldsymbol{\Omega}$ are related by $\boldsymbol{\Omega} = \boldsymbol{\Theta}_D^{-1/2}\boldsymbol{\Theta}$ and $\boldsymbol{\Theta} = \boldsymbol{\Omega}_D\boldsymbol{\Omega}$. Note $\boldsymbol{\Omega}$ may not be symmetric. However, the sparsity patterns of $\boldsymbol{\Omega}$ and $\boldsymbol{\Theta}$ coincide with each other, so the $\ell_1$ penalty on $\boldsymbol{\Omega}_{-D}$ is legitimate. Under some regulatory conditions, the estimate $\hat{\boldsymbol{\Omega}}$ that uniquely minimizes (3) is a consistent estimator of $\boldsymbol{\Omega}^* := \boldsymbol{\Theta}_D^{*-1/2}\boldsymbol{\Theta}^*$ and its sparsity pattern. The ACCORD framework lies in line with $\ell_1$-penalized pseudolikelihood-based methods for partial correlation network estimation. These methods are computionally more scalable than the graphical lasso owing to the use of pseudolikelihood, which in turn is more robust to non-normality. SPACE [29] is the first of its kind, which directly targets partial correlations. While SPLICE [32] and SYMLASSO [9] follow this line, these methods are not convex, and convergence of the corresponding optimization algorithms are not established. CONCORD [17] convexifies SPACE by changing the target variable to precision matrix $\boldsymbol{\Theta}$. Nevertheless, consistency of the estimated precision matrix requires quite strong conditions. ACCORD relaxes those by asymmetrizing the target variable and establishes non-asymptotic bounds on estimation error and sample complexity [19].

The key computational attraction of ACCORD is its scalability. In (3), the log-determinant barrier is applied only to the diagonal elements of $\boldsymbol{\Omega}$, hence reduces to merely a sum of logarithms. This makes the KKT conditions for (3) much more tractable than (2). In [19], a proximal gradient algorithm is proposed to solve (3) at a linear rate. The major computational bottleneck of this algorithm is sparse-dense matrix-matrix multiplication, which easily allows for distributed computation [18]. This contrasts ACCORD with the graphical lasso. For massive-scale data from, e.g., multi-omic studies, [19] implements the algorithm on distributed-memory high-performance computing (HPC) systems equipped with the message passing inferface (MPI). As a showcase, the liver hepatocellular carcinoma (LIHC) cohort from The Cancer Genome Atlas [1] with $p \approx 300,000$ molecular features, for which the most scalable graphical lasso algorithm [14] fails, are analyzed. Despite the success, this computation takes 200 node-hours on a powerful HPC system in a supercomputing center, accessibility to which is generally limited to public. For wider uses of ACCORD among a large number of scientists, a computational algorithm that can operate in a smaller but more accessible environment, such as a GPU workstation, and possesses a convergence rate faster than the linear rate of the proximal gradient algorithm (to compensate for the limited computing power) is on demand.

This paper addresses such a demand by proposing a semismooth Newton method for solving (3). Because of its superlinear convergence property, semismooth Newton methods [31] have been widely employed for learning problems with non-smooth objective functions [21, 23, 33, 35]. We develop the semismooth Newton method for ACCORD through two steps. First, we show that problem (3) is rowwise separable, which allows it to decompose into separate $p$-dimensional minimization problems for each row of $\boldsymbol{\Omega}$. Second, we show that the Karush-Kuhn-Tucker (KKT) condition for

each minimization problem can be formulated as finding a root of a nonlinear multivariate function $F$, which is nondifferentiable but strongly semismooth. This observation ensures that the iteration converges at a locally quadratic rate. The key to implementation is to find a suitable element of Clarke's generalized Jacobian (which is a set) of $F$. We achieve this goal by deriving the Bouligand derivative of $F$. The latter allows us to *globalize* the semismooth Newton algorithm, for which the initial point can be chosen with liberty. A connection to the primal-dual active set (PDAS) strategy [12] is also made. We then examine the performance of our algorithm in both synthetic and the LIHC data, with concluding remarks provided thereafter. Proofs of the results are provided in the Appendix.

## 2 Locally convergent semismooth Newton method

### 2.1 KKT condition as root-finding

In this subsection, we show that the KKT condition for (3) can be formulated as finding a root of a multivariate nonlinear function. We begin with observing that the objective function in (3) is separable in rows. Let $\boldsymbol{\Omega}_i$ be the $i$-th row of $\boldsymbol{\Omega}$. Then, (3) can be rewritten as finding a minimizer of

$$\sum\nolimits_{i=1}^{p} \left( -\log\det\boldsymbol{\Omega}_{i,i} + (1/2)\,\mathbf{tr}(\boldsymbol{\Omega}_i^T\boldsymbol{\Omega}_i\boldsymbol{S}) + \lambda\|\boldsymbol{\Omega}_{i,-i}\|_1 \right), \tag{4}$$

where $\boldsymbol{\Omega}_{i,i}$ and $\boldsymbol{\Omega}_{i,-i}$ denote the diagonal and the off-diagonal parts of $\boldsymbol{\Omega}_i$, respectively. Equation (4) clearly shows that the objective function is separable in $\boldsymbol{\Omega}_i$'s. Henceforth, we will derive the optimality condition for $i = 1$ without loss of generality. Let $\omega = (\omega_1, \omega_{-1}^\top)^\top$, where $\omega_1 = \boldsymbol{\Omega}_{1,1}$ and $\omega_{-1} = \boldsymbol{\Omega}_{1,-1}$. Then, the objective function to minimize is $f(\omega) = g(\omega) + h(\omega)$, where

$$g(\omega) = (1/2)\omega^T\boldsymbol{S}\omega, \quad h(\omega) = -\log\omega_1 + \lambda\|\omega_{-1}\|_1,$$

which are both convex functions. By introducing a dual variable $d = (d_1, d_{-1}^\top)^\top$, the optimality condition $\partial f(\omega^*) = 0$ can be expressed as

$$\boldsymbol{S}\omega^* + d^* = 0, \quad d^* \in \partial h(\omega^*) \tag{5}$$

where $\partial f$ and $\partial h$ denote the subdifferentials of $f$ and $h$, respectively. Moreover, the condition $d^* \in \partial h(\omega^*)$ can be equivalently expressed using the proximal operator of $h$ [15]

$$\omega^* = \mathbf{prox}_h(\omega^* + d^*), \tag{6}$$

which can be explicitly written as

$$\mathbf{prox}_h(\omega) := \arg\min_{v\in\mathbb{R}^p}\left\{h(v) + (1/2)\|v - \omega\|_2^2\right\} = \left(\tfrac{\omega_1+\sqrt{\omega_1^2+4}}{2},\ T_\lambda(\omega_{-1})^\top\right)^\top,$$

where $T_\lambda(\omega_{-1}) := (\text{sign}(\omega_j)\max(|\omega_j| - \lambda, 0))_{j\neq1}$ is an element-wise soft-thresholding operator. Combining (5) and (6), the KKT condition for (3) can be equivalently written as

$$F(\omega^*, d^*) = 0, \tag{7}$$

where $F : \mathbb{R}^{2p} \to \mathbb{R}^{2p}$ is defined as

$$F(\omega, d) = \begin{pmatrix} \boldsymbol{S}\omega + d \\ \omega_1 - \mathbf{prox}_{-\log(\cdot)}(\omega_1 + d_1) \\ \omega_{-1} - T_\lambda(\omega_{-1} + d_{-1}) \end{pmatrix}, \quad \mathbf{prox}_{-\log(\cdot)}(x) = \frac{x + \sqrt{x^2 + 4}}{2}. \tag{8}$$

Therefore, solving (3) amounts to finding a root of nonlinear equation (7). Map $F$ has a unique root almost surely, as long as the sample is drawn from a continuous distribution [19, Theorem 3.1].

### 2.2 Newton's method for semismooth functions

The formulation (7)–(8) of the KKT condition naturally suggests Newton's method for finding the root. However, because of the presence of the soft thresholding operator in (8), it is not differentiable everywhere, which means that the plain Newton iteration cannot be applied. In this section, we show that, despite the nondifferentiablity, function $F$ in (8) is strongly semismooth, hence a semismooth version of Newton's method can be employed. We begin by defining semismoothness [26, 31].

**Definition 2.1** (Semismoothness). Function $F : \mathbb{R}^m \to \mathbb{R}^l$ is said to be semismooth at $x \in \mathbb{R}^m$ if it is locally Lipschitz, all directional derivatives $F'(x; v) := \lim_{t \to 0} \frac{F(x+tv) - F(x)}{t}$ exist, and for any $G \in \partial F(x + v)$, the following holds:

$$Gv - F'(x; v) = o(\|v\|).$$

Furthermore, $F$ is strongly semismooth at $x$ if it is semismooth at $x$ and satisfies the condition for any $G \in \partial F(x + v)$,

$$Gv - F'(x; v) = O(\|v\|^2).$$

The $F$ is said to be (strongly) semismooth on $U \in \mathbb{R}^m$ if it is (strongly) semismooth at any $x \in U$.

**Theorem 2.2.** *The map $(\omega, d) \mapsto F(\omega, d)$ as defined in* (8) *is strongly semismooth.*

The semismooth Newton method [31] relies on the notion of the generalized Jacobian [3].

**Definition 2.3** (Generalized Jacobian). For a function $F : \mathbb{R}^m \to \mathbb{R}^l$ that is locally Lipschitz continuous around $x \in \mathbb{R}^m$, Clarke's generalized Jacobian at $x$ is defined as

$$\partial F(x) = \mathrm{conv} \left\{ \lim_k \nabla F(x^k) : x^k \to x, \ x^k \in D_F(x) \right\},$$

where $D_F(x) = \{y : F \text{ is differentiable at } y\}$, $\mathrm{conv}$ denotes the convex hull operation, and $\nabla F(y)$ represents the Jacobian of $F$ at $y$. We call an element of $\partial F(x)$ a generalized Jacobian of $F$ at $x$.

Thus for locally Lipschitz functions, a version of Newton's method for solving $F(x) = 0$ can be defined as

$$x^{k+1} = x^k - G(x^k)^{-1} F(x^k), \quad G(x^k) \in \partial F(x^k), \tag{9}$$

provided that $G(x^k)$ is nonsingular. If $F$ semismoothness, then the Newton iteration (9) exhibits a superlinear convergence for suitable initialization [31]. If it is strongly semismooth, then it converges at a locally quadratic rate:

**Theorem 2.4** ([31]). *Suppose that $x^*$ is a root of $F : \mathbb{R}^m \to \mathbb{R}^l$, and that $F$ is strongly semismooth in an open neighborhood $U$ containing $x^*$, with a generalized Jacobian $G(x) \in \partial F(x)$. If $G(x)$ is nonsingular for all $x \in U$ and the set $\{\|G(x)^{-1}\| : x \in U\}$ is bounded, then the semismooth Newton method* (9) *converges quadratically to $x^*$, provided that $\|x^0 - x^*\|$ is sufficiently small.*

To verify the assumptions of Theorem 2.4 for the target problem (7), we provide a sufficient condition.

**Proposition 2.5.** *Suppose $(\omega^*, d^*)$ satisfies equation* (7) *for $F$ defined in* (8). *Let $\bar{\mathcal{A}}^* = \{1\} \cup \{i \in \{2, \dots, p\} : |\omega_i^* + d_i^*| > \lambda\}$. If $X_{\bar{\mathcal{A}}^*}$, the submatrix of $X$ taking the columns indexed by $\bar{\mathcal{A}}^*$ has full column rank, then there exists a nonsingular $G(\omega, d) \in \partial F(\omega, d)$ for any $(\omega, d) \in \mathbb{R}^{2p}$ in a neighborhood of $(\omega^*, d^*)$.*

In particular, we may choose (see Appendix D for the derivation)

$$G(\omega, d) = \begin{pmatrix} S & I \\ I - J & -J \end{pmatrix}, \tag{10}$$

where $J = \mathrm{diag}(J_{ii})$ is a diagonal matrix with $J_{11} = \frac{1}{2} + \frac{\omega_1 + d_1}{2\sqrt{(\omega_1 + d_1)^2 + 4}}$ and $J_{ii} = 1$ if $|\omega_i + d_i| > \lambda$, $J_{ii} = 0$ if $|\omega_i + d_i| \le \lambda$ for $i = 2, \dots, p$. In this case, semismooth Newton iteration (9) is equivalent to the primal-dual active set algorithm [11, 7, 2, 15].

## 3  Globalizing semismooth Newton method

### 3.1  B-semismooth Newton method

In this section, we propose a globalization strategy for the semismooth Newton method (9). Although the quadratic convergence result (Theorem 2.4) is attractive, this result is only local and is silent on how to choose the initial point $x^0 = (\omega^0, d^0)$. An algorithm that converges from any initial point while maintaining locally quadratic convergence near the solution is more practical and desirable. To this end, it turns out that globalizing iteration (9) *per se* is difficult, while globalizing the semismooth Newton based on Bouligand derivatives or B-derivatives [28] is manageable.

**Definition 3.1** (B-differentiability). A function $F : \mathbb{R}^m \to \mathbb{R}^l$ is said Bouligand differentiable (B-differentiable) at $x \in \mathbb{R}^m$ if it is locally Lipschitz, the directional derivative $F'(x; v) := \lim_{t \downarrow 0} \frac{F(x+tv)-F(x)}{t}$ exists for all $v$ and fulfill the approximation property $\|F(x + v) - F(x) - F'(x; v)\| = o(\|v\|)$ as $v \to 0$. If $F$ is B-differentiable for any $x \in \mathbb{R}^m$, then it is said B-differentiable. The directional derivative that satisfies these conditions is called the Bouligand derivative (B-derivative) of $F$ at $x$ in the direction $v$.

Using the directional derivative $F'(x; v)$, a Newton method for finding the root of a B-differentiable function $F$ can defined as

$$x^{k+1} = x^k + v^k, \quad F'(x^k; v^k) = -F(x^k). \tag{11}$$

Our target function is B-differentiable:

**Proposition 3.2.** *Function $F$ as defined in* (8) *is B-differentiable. Moreover, its B-derivative at $(\omega, d)$ in the direction of $v = (v_\omega, v_d)$ is given by*

$$F'(\omega, d; v) = \begin{pmatrix} \boldsymbol{S} v_\omega + v_d \\ (z + 1/2)v_{\omega,1} + (z - 1/2)v_{d,1} \\ \begin{cases} -v_{d,i} & i \in \mathcal{A} \\ v_{\omega,i} & i \in \mathcal{I}^0 \\ \min\{v_{\omega,i}, -v_{d,i}\} & i \in \mathcal{I}^+ \\ \max\{v_{\omega,i}, -v_{d,i}\} & i \in \mathcal{I}^- \end{cases} \end{pmatrix}, \tag{12}$$

*where $z = z(\omega_1, d_1) := -\frac{\omega_1 + d_1}{2\sqrt{(\omega_1 + d_1)^2 + 4}}$ and $\mathcal{A} = \mathcal{A}(\omega, d)$, $\mathcal{I}^0 = \mathcal{I}^0(\omega, d)$, $\mathcal{I}^+ = \mathcal{I}^+(\omega, d)$, and $\mathcal{I}^- = \mathcal{I}^-(\omega, d)$ such that*

$$\begin{aligned} \mathcal{A} &= \{i \in \{2, \dots, p\} : |\omega_i + d_i| > \lambda\}, \quad \mathcal{I}^0 = \{i \in \{2, \dots, p\} : |\omega_i + d_i| < \lambda\}, \\ \mathcal{I}^+ &= \{i \in \{2, \dots, p\} : \omega_i + d_i = \lambda\}, \quad \mathcal{I}^- = \{i \in \{2, \dots, p\} : \omega_i + d_i = -\lambda\}. \end{aligned} \tag{13}$$

### 3.2 Solvability of the B-semismooth Newton method via linear complementarity problem

Applying (12) to (11), finding the Newton direction $v$ such that $F'(\omega, d; v) = -F(\omega, d)$ reduces to

$$\boldsymbol{S} v_\omega + v_d = -\boldsymbol{S}\omega - d \tag{14}$$
$$(z + 1/2)v_{\omega,1} + (z - 1/2)v_{d,1} = -\omega_1 + \mathbf{prox}_{-\log(\cdot)}(\omega_1 + d_1) \tag{15}$$
$$\min\{v_{\omega,i}, -v_{d,i}\} = -\omega_i, \ i \in \mathcal{I}^+, \ \max\{v_{\omega,i}, -v_{d,i}\} = -\omega_i, \ i \in \mathcal{I}^-, \tag{16}$$

where $z$ is the same as in (12). Determining the existence of a solution to (14)–(16) and finding one is a quadratic programming problem in $v = (v_\omega, v_d) \in \mathbb{R}^{2p}$. However, under the condition similar to that of Proposition 3.2, we only need to solve a much smaller linear complementarity problem (LCP) [4]. Let $\bar{\mathcal{A}} = \bar{\mathcal{A}}(\omega, d) := \{1\} \cup \mathcal{A}$ and $\bar{\bar{\mathcal{A}}} = \bar{\bar{\mathcal{A}}}(\omega, d) := \bar{\mathcal{A}} \cup \mathcal{I}^+ \cup \mathcal{I}^-$ from (13) and $\boldsymbol{X}_{\bar{\mathcal{A}}}$, $\boldsymbol{X}_{\bar{\bar{\mathcal{A}}}}$ be the submatrix of $\boldsymbol{X}$ obtained in a similar way to $\boldsymbol{X}_{\bar{\mathcal{A}}^*}$ in Proposition 2.5.

**Lemma 3.3** (cf. [10]). *Assume $\boldsymbol{X}_{\bar{\bar{\mathcal{A}}}}$ has full column rank. Given $(\omega, d) \in \mathbb{R}^{2p}$, $v = (v_\omega, v_d) \in \mathbb{R}^{2p}$ solves* (14)–(16) *if and only if*

$$\mathbf{x} = \left( (-v_{d, \mathcal{I}^+} + \omega_{\mathcal{I}^+})^\top, (v_{d, \mathcal{I}^-} - \omega_{\mathcal{I}^-})^\top \right)^\top, \ \mathbf{y} = \left( (v_{\omega, \mathcal{I}^+} + \omega_{\mathcal{I}^+})^\top, (-v_{\omega, \mathcal{I}^-} - \omega_{\mathcal{I}^-})^\top \right)^\top$$

*solves the LCP*

$$\mathbf{x} \geq \mathbf{0}, \ \mathbf{y} \geq \mathbf{0}, \ \langle \mathbf{x}, \mathbf{y} \rangle = 0, \ and \ \mathbf{y} = M\mathbf{x} + q \tag{17}$$

*where $M = M(\boldsymbol{S}, \omega, d)$ and $q = q(\boldsymbol{S}, \omega, d)$ are given in Appendix E.2. The $M$ is symmetric positive definite and* (17) *admits a unique solution.*

A LCP can be solved efficiently using standard techniques such as Lemke's or pivoting methods [4]. Let $S_{\mathcal{R}, \mathcal{C}}$ be the submatrix of $\boldsymbol{S}$ with rows indexed by $\mathcal{R}$ and columns indexed by $\mathcal{C}$. With the solution to (17) available, the Newton direction $v^k$ in (11) for the $F$ in (8) can be found in a closed form as stated in the following main theorem.

**Theorem 3.4.** *In* (13), *let* $\mathcal{A} = \mathcal{A}(\omega^k, d^k)$, $\bar{\bar{\mathcal{A}}} = \bar{\bar{\mathcal{A}}}(\omega^k, d^k)$, *and* $\mathcal{I}^{\pm} = \mathcal{I}^{\pm}(\omega^k, d^k)$. *Assume* $\boldsymbol{X}_{\bar{\mathcal{A}}}$ *has full column rank. Let* $\mathbf{x}, \mathbf{y}$ *be the unique solution to* (17) *for an iterate* $(\omega^k, d^k)$. *In addition to* (13), *let* $\mathcal{A}^+ = \{i \in \{2, \ldots, p\} : \omega_i^k + d_i^k > \lambda\}$ *and* $\mathcal{A}^- = \{i \in \{2, \ldots, p\} : \omega_i^k + d_i^k < -\lambda\}$. *Then, the B-semismooth Newton iteration* (11) *is well-defined with*

$$v_{\omega,\mathcal{I}^0}^k = -\omega_{\mathcal{I}^0}^k, \quad v_{d,\mathcal{A}}^k = \lambda(\mathbf{1}_{\mathcal{A}^+}^\top, -\mathbf{1}_{\mathcal{A}^-}^\top)^\top - d_{\mathcal{A}}^k, \quad \begin{pmatrix} v_{d,\mathcal{I}^+}^k \\ v_{d,\mathcal{I}^-}^k \end{pmatrix} = \begin{pmatrix} -\mathbf{x}_{\mathcal{I}^+} \\ \mathbf{x}_{\mathcal{I}^-} \end{pmatrix} + \begin{pmatrix} \omega_{\mathcal{I}^+}^k \\ \omega_{\mathcal{I}^-}^k \end{pmatrix},$$

$$\left((v_{\omega,1}^k)^\top, (v_{\omega,\mathcal{A}}^k)^\top, (v_{\omega,\mathcal{I}^+}^k)^\top, -(v_{\omega,\mathcal{I}^-}^k)^\top, (v_{d,\mathcal{I}^0}^k)^\top\right)^\top = N^{-1}\left(P^k - \left(0^\top, 0^\top, (v_{d,\mathcal{I}^+}^k)^\top, -(v_{d,\mathcal{I}^-}^k)^\top, 0^\top\right)^\top\right),$$

$$v_{d,1} = b^k v_{\omega,1}^k - c^k, \quad N^{-1} = \begin{pmatrix} N_1^{-1} & 0 \\ -CN_1^{-1} & I \end{pmatrix}$$

*where* $b^k, c^k \in \mathbb{R}$, $P^k \in \mathbb{R}^p$, $C \in \mathbb{R}^{(p-|\bar{\bar{\mathcal{A}}}|) \times |\bar{\bar{\mathcal{A}}}|}$, *and* $N_1 \in \mathbb{R}^{|\bar{\bar{\mathcal{A}}}| \times |\bar{\bar{\mathcal{A}}}|}$, *given in Appendix E.2, are determined by* $\omega^k, d^k$ *and* $\boldsymbol{S}$. *In particular,* $N_1$ *is symmetric positive definite.*

In particular, given that $\mathcal{I}^{+^*} \cup \mathcal{I}^{-^*} = \emptyset$ for $\mathcal{I}^{+^*} = \{i \in \{2, \ldots, p\} : \omega_i^* + d_i^* = \lambda\}$, $\mathcal{I}^{-^*} = \{i \in \{2, \ldots, p\} : \omega_i^* + d_i^* = -\lambda\}$, the convergence of (11) becomes locally quadratic:

**Corollary 3.5** ([30, Corollary 3.3]). *Let* $(\omega^*, d^*)$ *be the unique solution to* $F(\omega, d) = 0$. *If* $\mathcal{I}^{+^*} \cup \mathcal{I}^{-^*} = \emptyset$ *and* $\boldsymbol{X}_{\bar{\mathcal{A}}^*}$ *from Proposition 2.5 has full column rank, then the B-semismooth Newton iteration* (11) *is well-defined and converges quadratically to* $(\omega^*, d^*)$ *in its neighborhood.*

*Remark* 3.6. Multiplication with matrix $N^{-1}$ only requires a linear system solve involving the symmetric positive definite submatrix $N_1$ of size $|\bar{\bar{\mathcal{A}}}| \times |\bar{\bar{\mathcal{A}}}|$. In the sparse estimation context, the size of the active set $\bar{\bar{\mathcal{A}}}$ is small, hence solving (11) is not a bottleneck for scalability.

*Remark* 3.7. Given that $F$ in (8) is locally Lipschitz and directionally differentiable, there exists a generalized Jacobian $G$ such that $F'(\omega, d; v) = G(\omega, d)v$ where $v$ satisfies (11) [31]. More specifically, the B-semismooth Newton method can be regarded as the semismooth Newton method (9) that employs the generalized Jacobian $G$ in (10) such that $J_{ii} = 1$ for $i \in \bar{\mathcal{A}} \cup \{i \in \mathcal{I}^+ \cup \mathcal{I}^- : \mathbf{x}_i = 0\}$ and $J_{ii} = 0$ for $i \in \mathcal{I}^0 \cup \{i \in \mathcal{I}^+ \cup \mathcal{I}^- : \mathbf{x}_i > 0\}$, where $\mathbf{x}$ is the solution to the LCP (17). Furthermore, since it is numerically unlikely that $|\omega_i + d_i|$ precisely equals to $\lambda$, in most scenarios $\mathcal{I}^+ \cup \mathcal{I}^- = \emptyset$ and the B-semismooth Newton method coincides with the PDAS algorithm.

### 3.3 Globalizing B-semismooth Newton with line search

Globalization of the B-semismooth Newton method (11) is based on the observation that, under the (unique) existence of the solution to (7), solving this nonlinear equation is equivalent to minimizing $\theta(\omega, d) = \frac{1}{2}\|F(\omega, d)\|_2^2$. Given the Newton direction $v^k$ obtained by solving (14)–(16) for $(\omega, d) = (\omega^k, d^k)$, we can apply line search to construct a damped Newton algorithm that monotonically decreases the objective sequence $\{\theta(\omega^k, d^k)\}$. Following [25], we consider the line search algorithm that finds an integer $m_k$ such that

$$\theta(x^k + \rho^{m_k} v^k) - \theta(x^k) \le -2\sigma \rho^{m_k} \theta(x^k), \tag{18}$$

for algorithm parameters $\sigma, \rho \in (0, 1)$.

Also, Theorem 3.4 suggests that the full column rank of the submatrix $\boldsymbol{X}_{\bar{\mathcal{A}}_k}$ is essential in computing the direction $v^k$ for $\bar{\bar{\mathcal{A}}}_k = \bar{\bar{\mathcal{A}}}(\omega^k, d^k)$. If this condition is not met, then we can resort to the proximal gradient algorithm of [19] which converges linearly to the solution. Note that this algorithm does not necessarily descent $\theta(\omega, d)$ in one iteration. Thus, we need repeat the step for $\theta(\omega, d)$ to decrease.

The resulting damped B-semismooth Newton algorithm is summarized in Algorithm 1. The main question is whether the iterate sequence $\{(\omega^k, d^k)\}$ will converge to the solution $(\omega^*, d^*)$, and whether the Newton steps will eventually take place; these are all addressed in the following theorem.

**Theorem 3.8.** *Suppose* $\mathcal{I}^{+^*} \cup \mathcal{I}^{-^*} = \emptyset$ *and* $\boldsymbol{X}_{\bar{\mathcal{A}}^*}$ *from Proposition 2.5 has full column rank. Let* $\{(\omega^k, d^k)\}$ *be the sequence of iterates generated by Algorithm 1. Then, the proximal gradient step occurs only a finite number of times, and if we let* $\{\rho^{m_k}\}$ *denote the corresponding step sizes,*

    *1. if* $\limsup_{k \to \infty} \rho^{m_k} > 0$, *then* $(\omega^k, d^k) \to (\omega^*, d^*)$, *the unique solution to* $F(\omega, d) = 0$;

---

**Algorithm 1:** Damped B-semismooth Newton method with line search

---

**Input:** Initial guess $x^0 = (\omega^0, d^0)$ for all rows, $\rho_0, \sigma \in (0, 1)$, and tolerance *tol*

**1 foreach** *row in $\Omega$ simultaneously* **do**

**2** $\quad$ $k \leftarrow 0$;

**3** $\quad$ **while** $\theta(x^k) > tol$ **do**

**4** $\quad\quad$ Compute the sets $\mathcal{A}_k = \mathcal{A}(\omega^k, d^k)$, $\mathcal{I}_k^0 = \mathcal{I}^0(\omega^k, d^k)$, and $\mathcal{I}_k^\pm = \mathcal{I}^+(\omega^k, d^k) \cup \mathcal{I}^-(\omega^k, d^k)$;

**5** $\quad\quad$ Set $\bar{\bar{\mathcal{A}}}_k = \{i\} \cup \mathcal{A}_k \cup \mathcal{I}_k^\pm$;

**6** $\quad\quad$ **if** $X_{\bar{\bar{\mathcal{A}}}_k}$ *has full column rank* **then**

**7** $\quad\quad\quad$ Find the search direction $v^k$ according to Theorem 3.4;

**8** $\quad\quad\quad$ $m \leftarrow 0$;

**9** $\quad\quad\quad$ **repeat**

**10** $\quad\quad\quad\quad$ $m \leftarrow m + 1$;

**11** $\quad\quad\quad$ **until** $\theta(x^k + \rho_k^m v^k) \leq (1 - 2\sigma\rho_k^m)\theta(x^k)$;

**12** $\quad\quad\quad$ $x^{k+1} \leftarrow x^k + \rho_k^m v^k$;

**13** $\quad\quad$ **else**

**14** $\quad\quad\quad$ $\tilde{x} \leftarrow x^k$;

**15** $\quad\quad\quad$ **repeat**

**16** $\quad\quad\quad\quad$ Update $\tilde{x}$ with the proximal gradient algorithm of [19]

**17** $\quad\quad\quad$ **until** $\theta(\tilde{x}) \leq \theta(x^k)$;

**18** $\quad\quad\quad$ $x^{k+1} \leftarrow \tilde{x}$;

**19** $\quad\quad$ $k \leftarrow k + 1$;

---

2. *if* $\limsup_{k \to \infty} \rho^{m_k} = 0$ *and* $(\omega^\dagger, d^\dagger)$ *is an accumulation point of* $\{(\omega^k, d^k)\}$, *then* $(\omega^\dagger, d^\dagger) = (\omega^*, d^*)$ *and* $(\omega^k, d^k) \to (\omega^*, d^*)$.

*Moreover, if* $(\omega^k, d^k) \to (\omega^*, d^*)$, *then the convergence is locally quadratic.*

**Comparison to the SSNAL framework** There is a line of work that falls into the semismooth Newton-based augmented Lagrangian method (SSNAL) framework, developed for solving the lasso-related regression problems [21, 23, 35]. As the name suggests, this framework solves the problem via the inexact augmented Lagrangian method (ALM). ALM incurs an optimization subproblem, and SSNAL solves this subproblem via semismooth Newton. While the subproblem is solved at a locally superlinear (quadratic if it is strongly semismooth) rate, the convergence rate of the the outer ALM iteration is locally linear, even if the subproblem can be solved at a locally quadratic rate. Global convergence is also established.

In contrast, Algorithm 1 formulates the KKT condition as a strongly semismooth equation (8) and apply semismooth Newton. Globalization is achieved via damping instead of ALM. So our algorithmic design is fundamentally different from SSNAL: there is no subproblem to solve inexactly, and the convergence is locally quadratic. (If the KKT equation were merely semismooth, then the local convergence would be superlinear.)

Another distinction is that SSNAL typically constructs augmented Lagrangian via the *dual* of the original (primal) problem, while we focus on the primal. It is worth pointing out that [23] applies the SSNAL framework directly to the primal in addition to the dual, but its design and analysis focus on low-dimensional settings where the number of observations significantly exceeds the number of features (i.e., $n \gg p$). As such, it is not readily applicable to high-dimensional settings, which are the primary focus of our work.

## 4 Experiments

### 4.1 Convergence behavior of Damped B-semismooth Newton method

We present numerical experiments to empirically validate the convergence and descent property of the proposed damped B-semismooth Newton method. Specifically, we compare Algorithm 1 against the proximal gradient algorithm proposed in [19], by tracking the following three metrics over the

iteration:

$$\|F(\omega^k, d^k)\|_2^2, \quad \text{(objective function of Algorithm 1)}$$
$$f(\omega^k) - f(\hat{\omega}), \quad \text{(objective function of ACCORD)} \tag{19}$$
$$\|\omega^k - \omega^*\|_2^2. \quad \text{(squared distance to the optimum)}$$

The $\hat{\omega}$ estimates the optimal solution $\omega^*$ by the output of Algorithm 1 at convergence. Convergence is declared if all rows satisfies the criterion $\|F(\omega^k, d^k)\|^2 < 10^{-10}$. The rank test for line 6 in Algorithm 1 is based on the sufficiency check $|\bar{\bar{\mathcal{A}}}_k| \le n$, while more stringent tests (e.g., condition number or singular value thresholds) are also possible. For the line search parameter (line 11), we use $\rho_k = \max\{0.7 - 0.003 \cdot k, 0.4\}$ and $\sigma = 0.001$. The algorithm is implemented in PyTorch 1.13.1. Matrix operations are parallelized on six NVIDIA RTX 6000 Ada generation GPUs using CUDA 11.7.

To assess the convergence behavior, we generated $n = 500$ observations from a $p = 1,000$-dimensional zero-mean multivariate Gaussian distribution with precision matrix $\boldsymbol{\Theta}^*$ that contained 3% non-zero entries, locations of which were sampled uniformly at random. The regularization parameter $\lambda$ was set to 0.1, 0.15, and 0.2 to ensure that the estimated precision matrix exhibits a sparsity level comparable to $\boldsymbol{\Theta}^*$.

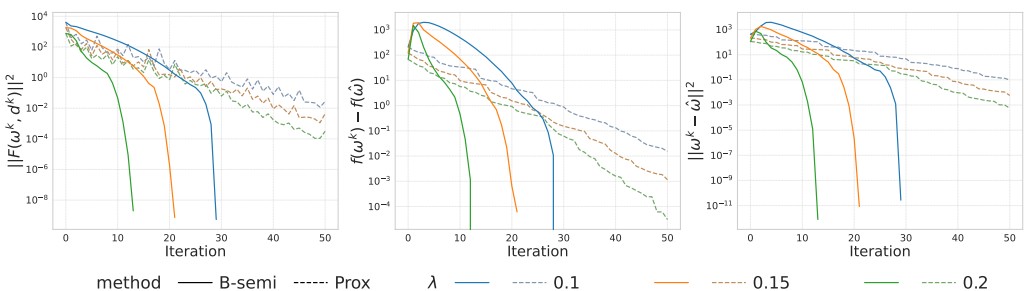

Figure 1: Convergence comparison: damped B-semismooth Newton vs. proximal gradient method

The results are summarized in Figure 1, where the vertical axis is plotted on a logarithmic scale. The proximal gradient method exhibits an approximately linear convergence rate with respect to both the ACCORD objective function $f$ and the squared distance to the optimum, consistent with the theory [19]. In contrast, the damped B-semismooth Newton method demonstrates superlinear convergence for both metrics, achieving approximately an order of magnitude faster convergence. Furthermore, for $\|F(\omega^k, d^k)\|^2$ in which Algorithm 1 has a descent property, the convergence appears locally quadratic (as predicted by theory) and occurs within a few tens of iterations.

As Algorithm 1 includes two components in addition to the semismooth Newton steps, namely the proximal gradient steps and the LCP solve step, we also conducted an ablation study to assess the contribution of each component. See Appendix G.1 for details.

## 4.2 Graph density-convergence trade-off

In Algorithm 1, the B-semismooth Newton update cannot be applied to the row in which $\text{rank}(\boldsymbol{X}_{\bar{\mathcal{A}}_k}) > \min(n, p)$. Since the size of the active set is inherently linked to the sparsity of the true precision matrix $\boldsymbol{\Theta}^*$ (hence that of $\boldsymbol{\Omega}^*$), we vary the regularization parameter $\lambda$ to study the robustness of the algorithm (in terms of the occurrence of the Newton step) to the density of the underlying partial correlation graph.

To conduct this investigation, we define the graph density as $\text{Density} = |E|/\binom{p}{2}$ where $|E|$ denotes the number of undirected edges and $p$ is the number of nodes in the graph implied by $\boldsymbol{\Theta}^*$. Based on this definition, we constructed a graph structure sampled uniformly at random with densities of 0.03, 0.05, and 0.10. Note that the graph density in Section 4.1 is roughly 0.03. The partial correlation matrix corresponding to the graph was constructed to have the minimum eigenvalues of at least 0.2, with all non-zero entries having absolute values no less than 0.1. Based on this matrix, we generate $n = 100$ observations from a $p = 1,000$-dimensional zero-mean multivariate Gaussian distribution with precision matrix $\boldsymbol{\Theta}^*$.

The hyperparameter $\lambda$ was chosen so that the estimated solution exhibits a sparsity level comparable to that of the true precision matrix $\Theta^*$, while the line search parameters were set to $\rho = 0.5, \sigma = 0.001$. We then assesses whether convergence is achieved within 300 iterations for each case. This limit of iterations was chosen because from Section 4.1 we see $B$-semismooth Newton method typically converges within a few tens of iterations. The results are presented in Table 1.

| Density $\setminus \lambda$ | 0.2 | 0.15 | 0.1 | 0.09 | 0.08 | 0.07 |
|---|---|---|---|---|---|---|
| 0.03 | Y | Y | – | – | – | – |
| 0.05 | – | Y | Y | – | – | – |
| 0.10 | – | – | Y | Y | Y | N |

Table 1: Convergence within 300 iterations for varying graph density and $\lambda$ (Yes / No)

| Density $\setminus \lambda$ | 0.07 | 0.06 | 0.05 |
|---|---|---|---|
| 0.10 | 9.1% | 97.7% | 100% |

Table 2: Proportion of rows updated by proximal method after 300 iterations

With one exception, Algorithm 1 converged within 300 iterations. The exception occurred when the underlying graph was the densest and the regularization parameter was the smallest. Usually the denser the underlying graph becomes, the smaller regularization parameter is required to recover its structure. To further investigate this phenomenon, we summarize in Table 2 the proportion of rows updated by the proximal gradient method after 300 iterations for smaller values of $\lambda$.

This result shows that there is a trade-off between the density of the underlying graph to estimate and the convergence speed of Algorithm 1. Nevertheless, for a consistent estimation of the support of the underlying graph, a sparsity assumption is essential, which is tied to the identifiability condition of Proposition 3.2 [7]. Under this condition, the Newton step occurs eventually. Adaptive selection of the regularization parameter $\lambda$ under the sparsity regime, e.g., continuation strategy [7, 15], is an interesting future direction of study.

Further experiments on other types of random graphs were also conducted, the results of which can be found in Appendix G.3.

## 4.3 Application to large-scale genomic data

In this subsection, we assess the performance of Algorithm 1 in estimating the partial correlation graph from a large-scale multi-omic dataset. Specifically, we apply Algorithm 1 and proximal gradient method (ACCORD-FBS) to the LIHC dataset from The Cancer Genome Atlas [1], consisting of $p = 305,471$ features including RNA transcription levels and the DNA methylation status of human genomes from $n = 365$ samples. Estimating partial correlations in such high-dimensional biological data can facilitate the identification and validation of epigenomic and transcriptomic regulatory mechanisms in human cells. This dataset was analyzed in [19], in which an HPC version of ACCORD-FBS was run on two national supercomputing centers for several hours. We emphasize that we employed our own implementation of ACCORD-FBS for the GPU workstation of the previous subsection, utilizing the row-wise separability.

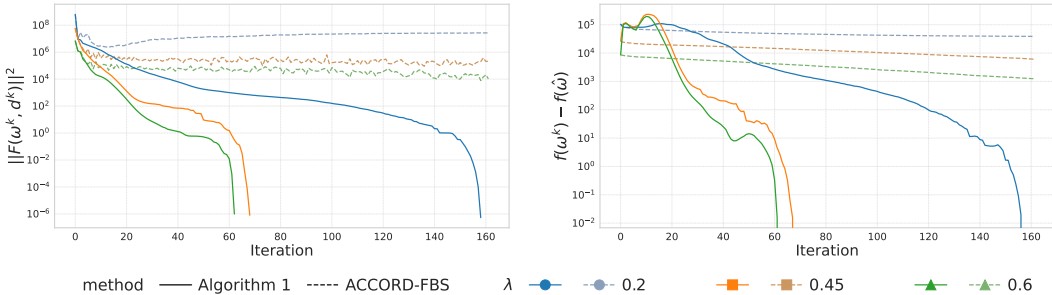

Figure 2: Convergence comparison on a large-scale multi-omic cancer dataset

The regularization parameter was set to $\lambda = 0.45$, selected via the extended pseudo-Bayesian information criterion [19]; a detailed sensitivity analysis of estimation performance around this value of $\lambda$ is provided in Appendix G.2. To test robustness of Algorithm 1, we further examined the optimization performance for $\lambda \in \{0.20, 0.60\}$, corresponding to relatively dense and sparse estimates. Figure 2

illustrates the convergence behavior of Algorithm 1 and ACCORD-FBS, evaluated using the first two metrics defined in (19). While ACCORD-FBS typically requires over a thousand iterations to converge [19], our algorithm reached the solution within only a few dozen iterations. Despite the reduced number of iterations, the computational cost per iteration remains nearly unchanged, as the dominant operation in each step involves only the inversion of a relatively small submatrix of dimension $|\bar{\bar{\mathcal{A}}}_k| \times |\bar{\bar{\mathcal{A}}}_k|$. Due to the rapid convergence of the B-semismooth Newton method, each iteration involves significantly fewer row updates compared to ACCORD-FBS.

| Method $\setminus \lambda$ | 0.2 | | 0.45 | | 0.6 | |
|---|---|---|---|---|---|---|
| | Time (s) | $\|F\|$ | Time (s) | $\|F\|$ | Time (s) | $\|F\|$ |
| Algorithm 1 | 5341 | $1.260 \times 10^1$ | 3352 | 0 | 2083 | 0 |
| ACCORD-FBS | 6778 | $4.709 \times 10^3$ | 5150 | $4.035 \times 10^2$ | 3452 | $2.671 \times 10^2$ |

Table 3: Wall-clock time (in seconds) and $\|F\|$ until convergence or 100 iterations

As shown in Table 3, Algorithm 1 consistently achieved shorter wall-clock time and smaller $\|F\|$ than ACCORD-FBS for all tested values of $\lambda$ within 100 iterations or at convergence, whichever came first.

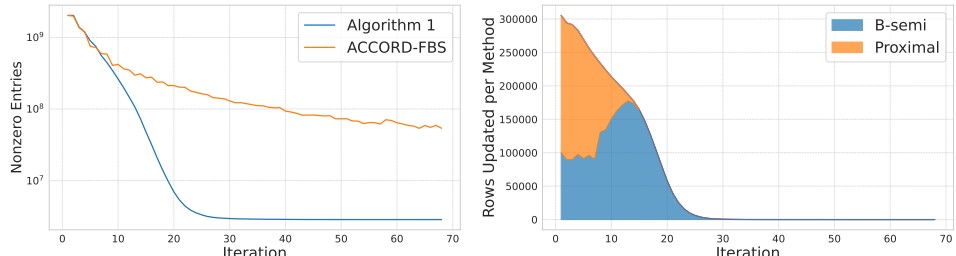

Figure 3: Interaction between sparsity and update strategy in Algorithm 1

As expected, more iterations were required to converge as the regularization parameter $\lambda$ decreases, and more likely the proximal gradient step occurs (rather than the Newton step) in Algorithm 1. As a means of dissecting this phenomenon, we fix $\lambda$ at 0.45 and monitor the number of nonzero elements in the iterate sequence $\mathbf{\Omega}^k$ and the number of rows updated by either of the two methods; see Figure 3. In the early stage, updates were performed primarily by the proximal gradient method. Over time, the number of nonzero entries in $\mathbf{\Omega}^k$ decreased and the proportion of rows updated by the Newton step increased. This transition aligns with intuition and enables Algorithm 1 to take advantage of the fast convergence behavior of the B-semismooth Newton method in later iterations.

## 5  Conclusion

We explored the applicability of semismooth Newton methods in partial correlation network estimation to derive a scalable algorithm with superlinear convergence. These methods exhibit locally quadratic convergence to the solution, where the global convergence is also achievable by combining with the proposed damping strategy. They also enjoy parallelism by leveraging the rowwise separability of the ACCORD objective function. Simulations and genomic data applications on GPUs validate the superior computational performance of the semismooth methods compared to the proximal method. In conclusion, semismooth Newton methods suggest a promising outlook for fast and efficient partial correlation analysis of large-scale data with modest computing resources.

## 6  Acknowledgements

S. Lee was partially supported by the Sejong Science Fellowship through the National Research Foundation of Korea (NRF) funded by the Ministry of Science and ICT (MSIT, No. RS-2024-00347617), Singapore Ministry of Education (MOE-000244-00, MOE-000617-00), and National Medical Research Council of Singapore (MOH-000986-00). D. Kim and J.-H. Won were supported

by the AI-Bio Research Grant through Seoul National University (No. 0413-20230050), Artificial Intelligence Graduate School Program (Seoul National University), and by the National Research Foundation of Korea (NRF) grant funded by the Korea government (MSIT, No. RS-2024-00337691).

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

# Supplementary Material for "Partial Correlation Network Estimation by Semismooth Newton Methods"

## A    Proof of Theorem 2.2

Recall that $F$ is given in (8),

$$F(\omega, d) = \begin{bmatrix} S\omega + d \\ \omega_1 - \mathbf{prox}_{-\log(\cdot)}(\omega_1 + d_1) \\ \omega_{-1} - T_\lambda(\omega_{-1} + d_{-1}) \end{bmatrix},$$

where $S$ is the sample covariance matrix, $T_\lambda$ is the soft-thresholding operator, and

$$\mathbf{prox}_{-\log(\cdot)}(x) = \frac{x + \sqrt{x^2 + 4}}{2}.$$

### A.1    F is locally Lipchitz

To prove the local Lipschitz continuity of $F$, it suffices to show that each component of $F$ is Lipschitz continuous. Note that each row of $S\omega + d$ is differentiable, with gradient $(S_i, 1)^T$, which norm is bounded by $\sqrt{\|S\|^2 + 1}$, and $\|S\|$ is a largest eigenvalue of $S$. $\omega_1 - \mathbf{prox}_{-\log(\cdot)}(\omega_1 + d_1)$ is also differentiable with gradient $(\frac{1}{2} - \frac{\omega_1 + d_1}{2\sqrt{(\omega_1 + d_1)^2 + 4}}, -\frac{1}{2} - \frac{\omega_1 + d_1}{2\sqrt{(\omega_1 + d_1)^2 + 4}})$, where the absolute value of each component are within $[0, 1]$. Finally, the soft-thresholding operator $T_\lambda(\omega_i + d_i)$ is Lipschitz continuous with a constant of $1$, so we can conclude that each component of $F$ is Lipschitz continuous.

### A.2    Directional differentiability of F

In here, we not only establish the directional differentiability of $F$, but also derive its directional derivative explicitly. A vector-valued function $F : \mathbb{R}^m \to \mathbb{R}^l$ is directionally differentiable at $x \in \mathbb{R}^m$ in the direction $v$ if and only if each of its components $F_i$ (for $i = 1, \ldots, l$) is directionally differentiable at $x$ in the direction $v$. Note that it is not necessary to consider the differentiable components, as the directional derivative of a differentiable function $F$ at a point $x$ in the direction $v$ is given by $F'(x; v) = \nabla F(x) \cdot v$. Accordingly, the directional derivative of the linear term $S\omega + d$ is simply $Sv_\omega + v_d$. Similarly, the directional derivative of the composite term $\omega_1 - \mathbf{prox}_{-\log(\cdot)}(\omega_1 + d_1)$ is $(z + 1/2)v_{\omega,1} + (z - 1/2)v_{d,1}$ where $z = -\frac{\omega_1 + d_1}{2\sqrt{(\omega_1 + d_1)^2 + 4}}$.

Now, we compute directional derivative of the non-differentiable component. We use the definition of active sets and inactive sets in (13). Then, the direction derivatives falls into one of the following cases:

i) $i \in \mathcal{I}^+(\omega, d)$ and $v_{\omega,i} + v_{d,i} > 0$. Then $i \in \mathcal{A}^+(\omega + tv_\omega, d + tv_d)$. In this case, $F(\omega_i + tv_{\omega,i}, d_i + tv_{d,i}) - F(\omega_i, d_i) = -tv_{d,i} + \lambda - \omega_i - d_i = -tv_{d,i}$. The same argument works for $i \in \mathcal{I}^-(\omega, d)$ and $v_{\omega,i} + v_{d,i} < 0$.

ii) $i \in \mathcal{I}^+(\omega, d)$ and $v_{\omega,i} + v_{d,i} < 0$. Then, $i \in \mathcal{I}^0(\omega + tv_\omega, d + tv_d)$. In this case, $F(\omega_i + tv_{\omega,i}, d_i + tv_{d,i}) - F(\omega_i, d_i) = \omega_i + tv_{\omega,i} - \omega_i = tv_{\omega,i}$. The same argument works for $i \in \mathcal{I}^-(\omega, d)$ and $v_{\omega,i} + v_{d,i} > 0$.

iii) $i \in \mathcal{A}(\omega, d)$. Then, we can find $t_0 > 0$ such that $\mathcal{A}(\omega, d) \subset \mathcal{A}(\omega + tv_\omega, d + tv_d)$ for all $0 < t \leq t_0$ with

$$t_0 \leq \min_{j \in \mathcal{A}(\omega, d)} \inf \{t > 0 : |\omega_j + tv_{\omega,j} + d_j + tv_{d,j}| > \lambda\}.$$

For $0 < t \leq t_0$ and $i \in \mathcal{A}(\omega, d) \subset \mathcal{A}(\omega + tv_\omega, d + tv_d)$,

$$F(\omega_i + tv_{\omega,i}, d_i + tv_{d,i}) - F(\omega_i, d_i) = -tv_{d,i}.$$

iv) $i \in \mathcal{I}^0(\omega, d)$. Then, we can find $t_1 > 0$ such that $\mathcal{I}^0(\omega, d) \subset \mathcal{I}^0(\omega + tv_\omega, d + tv_d)$ for all $0 < t \leq t_1$ with

$$t_1 \leq \min_{j \in \mathcal{I}^0(\omega, d)} \inf \{t > 0 : |\omega_j + tv_{\omega,j} + d_j + tv_{d,j}| < \lambda\}.$$

For $0 < t \leq t_1$ and $i \in \mathcal{I}^0(\omega, d) \subset \mathcal{I}^0(\omega + tv_\omega, d + tv_d)$,

$$F(\omega_i + tv_{\omega,i}, d_i + tv_{d,i}) - F(\omega_i, d_i) = tv_{\omega,i}.$$

Note that $t_0$ and $t_1$ are independent of $i$. Hence, directional derivative of $F$ is (12).

### A.3 Strong Semismoothness of F

Now, we prove that $F$ is strongly semismooth. From [31, Corollary 4.4], it is sufficient to check the component-wise strong semismoothness of $F$. First, since $\boldsymbol{S}\omega + d$ is an affine function, strong semismoothness holds trivially.

Next, we verify the strong semismoothness of $((\omega_1 - d_1) - \sqrt{(\omega_1 + d_1)^2 + 4})/2$. It suffices to consider only the non-linear term $H(\omega_1, d_1) := \sqrt{(\omega_1 + d_1)^2 + 4}$. The derivatives of $H(\omega_1, d_1)$ with respect to $\omega_1$ and $d_1$ are $\frac{\omega_1 + d_1}{\sqrt{(\omega_1 + d_1)^2 + 4}}$. The directional derivative of $H(\omega_1, d_1)$ in the direction of a vector $v = (v_\omega, v_d)^T$ is given by

$$H'(\omega_1, d_1; v_\omega, v_d) = \frac{\partial H}{\partial \omega_1} v_\omega + \frac{\partial H}{\partial d_1} v_d = \frac{\omega_1 + d_1}{\sqrt{(\omega_1 + d_1)^2 + 4}} (v_\omega + v_d).$$

and the generalized Jacobian of $H(\omega_1 + v_\omega, d_1 + v_d)$ is a matrix given by the partial derivatives with respect to $\omega_1$ and $d_1$ :

$$V = \partial H(\omega_1 + h_\omega, d_1 + h_d) = \left( \frac{\omega_1 + h_1 + d_1 + h_2}{\sqrt{(\omega_1 + h_1 + d_1 + h_2)^2 + 4}} \quad \frac{\omega_1 + h_1 + d_1 + h_2}{\sqrt{(\omega_1 + h_1 + d_1 + h_2)^2 + 4}} \right).$$

Then,

$$Vh - H'(\omega_1, d_1; h_\omega, h_d) = \left( \frac{\omega_1 + h_\omega + d_1 + h_d}{\sqrt{(\omega_1 + h_\omega + d_1 + h_d)^2 + 4}} - \frac{\omega_1 + d_1}{\sqrt{(\omega_1 + d_1)^2 + 4}} \right)(h_\omega + h_d).$$

Using a Taylor expansion, we obtain $\frac{\omega_1 + h_\omega + d_1 + h_d}{\sqrt{(\omega_1 + h_\omega + d_1 + h_d)^2 + 4}} - \frac{\omega_1 + d_1}{\sqrt{(\omega_1 + d_1)^2 + 4}} = H(\omega_1 + h_\omega, d_1 + h_d) - H(\omega_1, d_1) = \mathcal{O}(\|h\|)$. Thus, $Vh - H'(\omega_1, d_1; h_\omega, h_d) = \mathcal{O}(\|h\|^2)$.

Lastly, we have the facts that the soft thresholding operator is strongly semismooth everywhere [16] and $\omega \to \omega + d$ does not introduce any discontinuity or nonlinear behavior in $T_\lambda(\omega)$. Therefore, $\omega_{-1} - T_\lambda(\omega_{-1} + d_{-1})$ is strongly semismooth.

Since all components of $F$ are strongly semismooth, it follows that $F$ is strongly semismooth.

## B  Proof of Theorem 2.4

Before proving Theorem 2.4, we present a well-known condition that are equivalent to semismoothness from [11, Theorem 2.9, Theorem 2.12].

**Proposition B.1.** *Let $F : U \to \mathbb{R}^m$ be defined on the open set $U \subset \mathbb{R}^n$. Then, for $x \in U$, the following statements are equivalent:*

(a) *$F$ is semismooth at $x$.*

(b) *$F$ is locally Lipschitz continuous at $x$, all directional derivatives $F'(x; v)$ exist, and for any $G \in \partial F(x + v)$,*

$$\|F(x + v) - F(x) - Gv\| = o(\|v\|) \quad \text{as } v \to 0.$$

*Furthermore, the following statements are equivalent:*

(c) *$F$ is strongly semismooth at $x$.*

(d) *$F$ is locally Lipschitz continuous at $x$, all directional derivatives $F'(x; v)$ exist, and for any $G \in \partial F(x + v)$,*

$$\|F(x + v) - F(x) - Gv\| = \mathcal{O}(\|v\|^2) \quad \text{as } v \to 0.$$

Note that the Newton iterates satisfy

$$\|x^{k+1} - x^*\| \leq \|G(x^k)^{-1}\|\|F(x^k) - F(x^*) - G(x^k)(x^k - x^*)\|, \tag{20}$$

provided that $x^k \in U$. Let $B(x^*, r)$ denote a ball of radius $r$ centered at $x^*$ contained in $U$, and let $m$ be such that $\|G(x)^{-1}\| \leq m$ for all $x \in B(x^*, r)$. Assume that $F$ is semismooth at $x^*$ and let $\eta \in (0, 1]$ be arbitrary. Then, there exists $\rho \in (0, r)$ such that

$$\|F(x^* + v) - F(x^*) - G(x^* + v)v\| \leq \frac{\eta}{m}\|v\| \leq \frac{1}{m}\|v\| \tag{21}$$

for all $\|v\| < \rho$. Consequently, if we choose $x^0$ such that $\|x^0 - x^*\| < \rho$, then by induction from (20) and (21), for $v = x^k - x^*$, we have $\|x^{k+1} - x^*\| < \rho$ and $x^{k+1} \in B(x^*, \rho)$. It follows that the iterates are well-defined.

Assume further that $F$ is strongly semismooth at $x^*$, meaning there exists a constant $c > 0$ such that

$$\|F(x^* + v) - F(x^*) - G(x^* + v)v\| \leq c\|v\|^2 = \frac{c \cdot m}{m}\|v\|^2, \tag{22}$$

for all $\|v\| < \rho$.

Then, if we choose $x^0$ such that $\|x^0 - x^*\| < \rho$, then by induction from (20) and (22), for $v = x^k - x^*$, we have

$$\|x^{k+1} - x^*\| \leq m\|F(x^k) - F(x^*) - G(x^k)(x^k - x^*)\| \leq c \cdot m\|x^k - x^*\|^2,$$

where $\|x^{k+1} - x^*\| < \rho$. Therefore, the iterates remain well-defined and satisfy

$$\|x^{k+1} - x^*\|/\|x^k - x^*\|^2 = O(1),$$

indicating quadratic convergence of the sequence $\{x^k\}$ to $x^*$.

Thus, under the assumption of strong semismoothness of $F$, the Newton iterates converge quadratically to $x^*$. $\qquad\square$

## C   Proof of Proposition 2.5

Let

$$\mathcal{A} = \mathcal{A}(\omega, d) := \{i \in \{2, \ldots, p\} : |\omega_i + d_i| > \lambda\},$$
$$\mathcal{I} = \mathcal{I}(\omega, d) := \{i \in \{2, \cdots, p\} : |\omega_i + d_i| \leq \lambda\}.$$

Since $F$ is continuous, there exists a neighborhood $\mathcal{N}$ of $(\omega^*, d^*)$ such that for all $(\omega, d) \in \mathcal{N}$, we have $\bar{\mathcal{A}} = \{1\} \cup \mathcal{A} = \{1\} \cup \mathcal{A}^* = \bar{\mathcal{A}}^*$. Since $X_{\bar{\mathcal{A}}^*}$ has full column rank, $X_{\bar{\mathcal{A}}}$ also has full column rank for all $(\omega, d)$ sufficiently close to $(\omega^*, d^*)$.

One of the generalized Jacobian $G(\omega, d) \in \partial F(\omega, d)$ can be expressed as

$$G(\omega, d) = \begin{pmatrix} X_1^T X_1 & X_1^T X_{\mathcal{A} \cup \mathcal{I}} & 1 & 0 \\ X_{\mathcal{A} \cup \mathcal{I}}^T X_1 & X_{\mathcal{A} \cup \mathcal{I}}^T X_{\mathcal{A} \cup \mathcal{I}} & 0 & I \\ z + \frac{1}{2} & 0 & z - \frac{1}{2} & 0 \\ 0 & I_{\mathcal{I}} & 0 & -I_{\mathcal{A}} \end{pmatrix}$$

where $z = -\frac{\omega_1 + d_1}{2\sqrt{(\omega_1 + d_1)^2 + 4}}$. Note that we can permute the rows and columns of $G$ as

$$G(\omega, d) = \begin{pmatrix} \frac{1}{n} X_{\bar{\mathcal{A}}}^T X_{\bar{\mathcal{A}}} & \frac{1}{n} X_{\bar{\mathcal{A}}}^T X_{\mathcal{I}} & I_{\bar{\mathcal{A}}} & 0 \\ \frac{1}{n} X_{\mathcal{I}}^T X_{\bar{\mathcal{A}}} & \frac{1}{n} X_{\mathcal{I}}^T X_{\mathcal{I}} & 0 & I_{\mathcal{I}} \\ ((z + \frac{1}{2})_1 + 0_{\mathcal{A}}) & 0 & ((z - \frac{1}{2})_1 - I_{\mathcal{A}}) & 0 \\ 0 & I_{\mathcal{I}} & 0 & 0_{\mathcal{I}} \end{pmatrix}.$$

where $(x_1 + X_{\mathcal{A}}) = \begin{pmatrix} x_1 & 0 \\ 0 & X_{\mathcal{A}} \end{pmatrix}$.

Let $G := \begin{pmatrix} M_1 & M_2 \\ M_3 & M_4 \end{pmatrix}$, where $M_1 = \left(\frac{1}{n} X_{\bar{\mathcal{A}}}^T X_{\bar{\mathcal{A}}}\right)$, $M_2 = \left(\frac{1}{n} X_{\mathcal{I}}^T X_{\bar{\mathcal{A}}} \quad I_{\bar{\mathcal{A}}} \quad 0\right)$,

$$M_3 = \begin{pmatrix} \frac{1}{n}X_{\bar{\mathcal{A}}}^T X_{\mathcal{I}} \\ ((z+\frac{1}{2})_1 + 0_{\mathcal{A}}) \\ 0 \end{pmatrix} \text{ and } M_4 = \begin{pmatrix} \frac{1}{n}X_{\mathcal{I}}^T X_{\mathcal{I}} & 0 & I_{\mathcal{I}} \\ 0 & ((z-\frac{1}{2})_1 - I_{\mathcal{A}}) & 0 \\ I_{\mathcal{I}} & 0 & 0_{\mathcal{I}} \end{pmatrix}.$$

Note that $\left((z-\frac{1}{2})_1 - I_{\mathcal{A}}\right)^{-1}$ is $\left((1/(z-\frac{1}{2}))_1 - I_{\mathcal{A}}\right)$ and

$$\begin{pmatrix} \frac{1}{n}X_{\mathcal{I}}^T X_{\mathcal{I}} & 0 & I_{\mathcal{I}} \\ 0 & ((z-\frac{1}{2})_1 - I_{\mathcal{A}}) & 0 \\ I_{\mathcal{I}} & 0 & 0_{\mathcal{I}} \end{pmatrix} \begin{pmatrix} 0_{\mathcal{I}} & 0 & I_{\mathcal{I}} \\ 0 & ((1/(z-\frac{1}{2}))_1 - I_{\mathcal{A}}) & 0 \\ I_{\mathcal{I}} & 0 & -\frac{1}{n}X_{\mathcal{I}}^T X_{\mathcal{I}} \end{pmatrix} = I.$$

We utilize Schur complement of $G$ to formulate the inverse of $G$. Note that

$$G/M_4 = M_1 - M_2 M_4^{-1} M_3$$

$$= \frac{1}{n}X_{\bar{\mathcal{A}}}^T X_{\bar{\mathcal{A}}} - \begin{pmatrix} \frac{1}{n}X_{\mathcal{I}}^T X_{\bar{\mathcal{A}}} & I_{\bar{\mathcal{A}}} & 0 \end{pmatrix}$$

$$\cdot \begin{pmatrix} 0_{\mathcal{I}} & 0 & I_{\mathcal{I}} \\ 0 & ((1/(z-\frac{1}{2}))_1 - I_{\mathcal{A}}) & 0 \\ I_{\mathcal{I}} & 0 & -\frac{1}{n}X_{\mathcal{I}}^T X_{\mathcal{I}} \end{pmatrix} \begin{pmatrix} \frac{1}{n}X_{\bar{\mathcal{A}}}^T X_{\mathcal{I}} \\ ((z+\frac{1}{2})_1 + 0_{\mathcal{A}}) \\ 0 \end{pmatrix}$$

$$= \frac{1}{n}X_{\bar{\mathcal{A}}}^T X_{\bar{\mathcal{A}}} - \left((\tfrac{z+\frac{1}{2}}{z-\frac{1}{2}})_1 + 0_{\mathcal{A}}\right).$$

Given that $\left(X_{\bar{\mathcal{A}}}^T X_{\bar{\mathcal{A}}}\right)$ is nonsingular, $\left(X_{\bar{\mathcal{A}}}^T X_{\bar{\mathcal{A}}}\right) - n\left((\tfrac{z+\frac{1}{2}}{z-\frac{1}{2}})_1 + 0_{\mathcal{A}}\right)$ is also nonsingular since $|z| < \frac{1}{2}$. Thus, we can construct the inverse of $G$ as

$$G^{-1}(x) = \begin{pmatrix} (G/M_4)^{-1} & -(G/M_4)^{-1} M_2 M_4^{-1} \\ -M_4^{-1} M_3 (G/M_4)^{-1} & M_4^{-1} + M_4^{-1} M_3 (G/M_4)^{-1} M_2 M_4^{-1} \end{pmatrix}.$$

$\square$

# D    Derivation of a Generalized Jacobian for $F$

Let $F : \mathbb{R}^{2p} \to \mathbb{R}^{2p}$ be

$$F(\omega, d) = \begin{pmatrix} \mathbf{S}\omega + d \\ \omega_1 - \text{prox}_{-\log(\cdot)}(\omega_1 + d_1) \\ \omega_{-1} - T_\lambda(\omega_{-1} + d_{-1}) \end{pmatrix}, \quad \text{prox}_{-\log(\cdot)}(x) = \frac{x + \sqrt{x^2 + 4}}{2}$$

where $T_\lambda(\cdot)$ is the componentwise soft-thresholding operator.

Let $F_1(\omega, d) = \mathbf{S}\omega + d$ and $F_2(\omega, d) = \omega - \begin{pmatrix} \text{prox}_{-\log(\cdot)}(\omega_1 + d_1) \\ T_\lambda(\omega_{-1} + d_{-1}) \end{pmatrix}$.

Then, it is straightforward to see that $\frac{\partial F_1}{\partial \omega} = \mathbf{S}$, $\frac{\partial F_1}{\partial d} = \mathbf{I}$ and

$$\frac{\partial \text{prox}_{-\log(\cdot)}(\omega_1 + d_1)}{\partial \omega_1} = \frac{\partial \text{prox}_{-\log(\cdot)}(\omega_1 + d_1)}{\partial d_1} = \frac{1}{2} + \frac{\omega_1 + d_1}{2\sqrt{(\omega_1 + d_1)^2 + 4}}.$$

Note that for $i = 2, \cdots, p$,

$$\frac{\partial T_\lambda(\omega_i + d_i)}{\partial \omega_i} = \frac{\partial T_\lambda(\omega_i + d_i)}{\partial d_i} = \begin{cases} 0 & |\omega_i + d_i| > \lambda \\ [0,1] & |\omega_i + d_i| = \lambda \\ 1 & |\omega_i + d_i| < \lambda \end{cases}.$$

By adopting the convention $\frac{\partial T_\lambda(\omega_i + d_i)}{\partial \omega_i} = \frac{\partial T_\lambda(\omega_i + d_i)}{\partial d_i} = 0$ at the kink $\{|\omega_i + d_i| = \lambda\}$, the Jacobians simplify to $\frac{\partial F_2}{\partial \omega} = \mathbf{I} - \mathbf{J}$ and $\frac{\partial F_2}{\partial d} = -\mathbf{J}$ where $\mathbf{J} = \text{diag}(J_{ii})$ with

$$J_{11} = \frac{1}{2} + \frac{\omega_1 + d_1}{2\sqrt{(\omega_1 + d_1)^2 + 4}}, \quad J_{ii} = \begin{cases} 1 & |\omega_i + d_i| > \lambda \\ 0 & |\omega_i + d_i| \le \lambda, \end{cases} \quad i = 2, \ldots, p.$$

Thus,

$$\partial F(\omega, d) = \begin{pmatrix} \frac{\partial F_1}{\partial \omega} & \frac{\partial F_1}{\partial d} \\ \frac{\partial F_2}{\partial \omega} & \frac{\partial F_2}{\partial d} \end{pmatrix} = \begin{pmatrix} \mathbf{S} & \mathbf{I} \\ \mathbf{I} - \mathbf{J} & -\mathbf{J} \end{pmatrix},$$

which is equivalent to the example of the generalize Jacobian (10).

# E  Proofs of Theorem 3.4

## E.1  Proof of Proposition 3.2

In Appendix A.2, we established more than the existence of the directional derivative; we demonstrated a stronger property that $F(x + tv) - F(x) - tF'(x; v)$ is exactly zero for sufficiently small $t > 0$. Specifically, the main result in A.2 can be summarized as follows.

Let $(\omega, d) \in \mathbb{R}^{2p}$. There exists $s > 0$ such that, for all $0 < t \leq s$,

$$\frac{F(\omega + tv_\omega, d + tv_d) - F(\omega, d)}{t} = \begin{pmatrix} \boldsymbol{S}v_\omega + v_d \\ (z + 1/2)v_{\omega,1} + (z - 1/2)v_{d,1} \\ \begin{cases} -v_{d,i} & i \in \mathcal{A} \\ v_{\omega,i} & i \in \mathcal{I}^0 \\ \min\{v_{\omega,i}, -v_{d,i}\} & i \in \mathcal{I}^+ \\ \max\{v_{\omega,i}, -v_{d,i}\} & i \in \mathcal{I}^- \end{cases} \end{pmatrix}.$$

The right-hand side defines the component-wise directional derivative $F'(\omega, d; v)$ and $F'(\omega, d; \lambda v) = \lambda F'(\omega, d; v)$ for $\lambda \geq 0$. Consequently, $F$ satisfies the approximation property $\|F(x + v) - F(x) - F'(x; v)\| = o(\|v\|)$ as $v \to 0$. Since $F$ is locally Lipschitz (see Appendix A.1) and the directional derivative exists at every $v$, we conclude that $F$ is B-differentiable. $\square$

## E.2  Proofs of Lemma 3.3 and Theorem 3.4

**Proof of Lemma 3.3**: For the "only if" part, suppose $v = (v_\omega, v_d) \in \mathbb{R}^{2p}$ satisfies (14)–(16). We then observe that (16) is equivalent to

$$(v_{\omega,i} + \omega_i > 0 \cap -v_{d,i} + \omega_i = 0) \cup (v_{\omega,i} + \omega_i = 0 \cap -v_{d,i} + \omega_i \geq 0),$$
$$(v_{\omega,i} + \omega_i < 0 \cap -v_{d,i} + \omega_i = 0) \cup (v_{\omega,i} + \omega_i = 0 \cap -v_{d,i} + \omega_i \leq 0),$$

which in turn is equivalent to

$$v_{\omega,\mathcal{I}^+} + \omega_{\mathcal{I}^+} \geq 0, \quad -v_{d,\mathcal{I}^+} + \omega_{\mathcal{I}^+} \geq 0, \quad \langle v_{\omega,\mathcal{I}^+} + \omega_{\mathcal{I}^+}, -v_{d,\mathcal{I}^+} + \omega_{\mathcal{I}^+} \rangle = 0,$$
$$v_{\omega,\mathcal{I}^-} + \omega_{\mathcal{I}^-} \leq 0, \quad -v_{d,\mathcal{I}^-} + \omega_{\mathcal{I}^-} \leq 0, \quad \langle v_{\omega,\mathcal{I}^-} + \omega_{\mathcal{I}^-}, -v_{d,\mathcal{I}^-} + \omega_{\mathcal{I}^-} \rangle = 0. \tag{23}$$

Rewrite (14)–(15) as

$$\boldsymbol{S}v_\omega + v_d = -\boldsymbol{S}\omega - d,$$
$$(z + 1/2)v_{\omega,1} + (z - 1/2)v_{d,1} = -\frac{1}{2}\left((\omega_1 - d_1) - \sqrt{(\omega_1 + d_1)^2 + 4}\right),$$

where $z = z(\omega_1, d_1) = -\frac{\omega_1 + d_1}{2\sqrt{(\omega_1 + d_1)^2 + 4}}$.

The second equation can be equivalently written as

$$v_{d,1} = bv_{\omega,1} + c, \tag{24}$$

where

$$b = -\left(z - \frac{1}{2}\right)^{-1}\left(z + \frac{1}{2}\right) > 0, \quad c = -\frac{1}{2}\left(z - \frac{1}{2}\right)^{-1}\left((\omega_1 - d_1) - \sqrt{(\omega_1 + d_1)^2 + 4}\right). \tag{25}$$

Let $E = -\boldsymbol{S}\omega - d$ and $S_{\mathcal{R},\mathcal{C}}$ denote the submatrix of $\boldsymbol{S}$ with rows indexed by $\mathcal{R}$ and columns indexed by $\mathcal{C}$. Then, for the first equation,

$$\boldsymbol{S}v_\omega + v_d = E \iff \begin{pmatrix} S_{1,1}v_{\omega,1} + S_{1,2}v_{\omega,2} + \cdots + S_{1,p}v_{\omega,p} + v_{d,1} \\ S_{2,1}v_{\omega,1} + S_{2,2}v_{\omega,2} + \cdots + S_{1,p}v_{\omega,p} + v_{d,2} \\ \vdots \\ S_{p,1}v_{\omega,1} + S_{p,2}v_{\omega,2} + \cdots + S_{p,p}v_{\omega,p} + v_{d,p} \end{pmatrix} = \begin{pmatrix} E_1 \\ E_2 \\ \vdots \\ E_p \end{pmatrix}$$

$$\iff \begin{pmatrix} (S_{1,1} + b)v_{\omega,1} + S_{1,2}v_{\omega,2} + \cdots + S_{1,p}v_{\omega,p} \\ S_{2,1}v_{\omega,1} + S_{2,2}v_{\omega,2} + \cdots + S_{1,p}v_{\omega,p} + v_{d,2} \\ \vdots \\ S_{p,1}v_{\omega,1} + S_{p,2}v_{\omega,2} + \cdots + S_{p,p}v_{\omega,p} + v_{d,p} \end{pmatrix} = \begin{pmatrix} E_1 - c \\ E_2 \\ \vdots \\ E_p \end{pmatrix}.$$

Invoking the definition of $\mathcal{I}$ in (12), we can rearrange this equation as

$$
\begin{pmatrix}
(S_{1,1}+b)v_{\omega,1} + S_{1,-1}^T v_{\omega,-1} \\
S_{\mathcal{A},1}v_{\omega,1} + S_{\mathcal{A},-1}^T v_{\omega,-1} + v_{d,\mathcal{A}} \\
S_{\mathcal{I}^+,1}v_{\omega,1} + S_{\mathcal{I}^+,-1}^T v_{\omega,-1} + v_{d,\mathcal{I}^+} \\
S_{\mathcal{I}^-,1}v_{\omega,1} + S_{\mathcal{I}^-,-1}^T v_{\omega,-1} + v_{d,\mathcal{I}^-} \\
S_{\mathcal{I}^0,1}v_{\omega,1} + S_{\mathcal{I}^0,-1}^T v_{\omega,-1} + v_{d,\mathcal{I}^0}
\end{pmatrix}
=
\begin{pmatrix}
E_1 - c \\
E_{\mathcal{A}} \\
E_{\mathcal{I}^+} \\
E_{\mathcal{I}^-} \\
E_{\mathcal{I}^0}
\end{pmatrix}.
$$

In the above system, transpose the known $v_{\omega,\mathcal{I}^0}$ and $v_{d,\mathcal{A}}$ (from (8) and (12)) to the right-hand side of the equation to yield

$$
\begin{pmatrix}
(S_{1,1}+b)v_{\omega,1} + S_{1,\mathcal{A}\cup\mathcal{I}^+\cup\mathcal{I}^-}^T v_{\omega,\mathcal{A}\cup\mathcal{I}^+\cup\mathcal{I}^-} \\
S_{\mathcal{A},1}v_{\omega,1} + S_{\mathcal{A},\mathcal{A}\cup\mathcal{I}^+\cup\mathcal{I}^-}^T v_{\omega,\mathcal{A}\cup\mathcal{I}^+\cup\mathcal{I}^-} \\
S_{\mathcal{I}^+,1}v_{\omega,1} + S_{\mathcal{I}^+,\mathcal{A}\cup\mathcal{I}^+\cup\mathcal{I}^-}^T v_{\omega,\mathcal{A}\cup\mathcal{I}^+\cup\mathcal{I}^-} + v_{d,\mathcal{I}^+} \\
S_{\mathcal{I}^-,1}v_{\omega,1} + S_{\mathcal{I}^-,\mathcal{A}\cup\mathcal{I}^+\cup\mathcal{I}^-}^T v_{\omega,\mathcal{A}\cup\mathcal{I}^+\cup\mathcal{I}^-} + v_{d,\mathcal{I}^-} \\
S_{\mathcal{I}^0,1}v_{\omega,1} + S_{\mathcal{I}^0,\mathcal{A}\cup\mathcal{I}^+\cup\mathcal{I}^-}^T v_{\omega,\mathcal{A}\cup\mathcal{I}^+\cup\mathcal{I}^-} + v_{d,\mathcal{I}^0}
\end{pmatrix}
=
\begin{pmatrix}
E_1 - S_{1,\mathcal{I}^0}^T v_{\omega,\mathcal{I}^0} - c \\
E_{\mathcal{A}} - S_{\mathcal{A},\mathcal{I}^0}^T v_{\omega,\mathcal{I}^0} - v_{d,\mathcal{A}} \\
E_{\mathcal{I}^+} - S_{\mathcal{I}^+,\mathcal{I}^0}^T v_{\omega,\mathcal{I}^0} \\
E_{\mathcal{I}^-} - S_{\mathcal{I}^-,\mathcal{I}^0}^T v_{\omega,\mathcal{I}^0} \\
E_{\mathcal{I}^0} - S_{\mathcal{I}^0,\mathcal{I}^0}^T v_{\omega,\mathcal{I}^0}
\end{pmatrix}.
$$

Now, rearranging the above equation gives the following.

$$
\underbrace{\begin{pmatrix}
S_{1,1}+b & S_{1,\mathcal{A}} & S_{1,\mathcal{I}^+} & -S_{1,\mathcal{I}^-} & 0_{1,\mathcal{I}^0} \\
S_{\mathcal{A},1} & S_{\mathcal{A},\mathcal{A}} & S_{\mathcal{A},\mathcal{I}^+} & -S_{\mathcal{A},\mathcal{I}^-} & 0_{\mathcal{A},\mathcal{I}^0} \\
S_{\mathcal{I}^+,1} & S_{\mathcal{I}^+,\mathcal{A}} & S_{\mathcal{I}^+,\mathcal{I}^+} & -S_{\mathcal{I}^+,\mathcal{I}^-} & 0_{\mathcal{I}^+,\mathcal{I}^0} \\
-S_{\mathcal{I}^-,1} & -S_{\mathcal{I}^-,\mathcal{A}} & -S_{\mathcal{I}^-,\mathcal{I}^+} & S_{\mathcal{I}^-,\mathcal{I}^-} & 0_{\mathcal{I}^-,\mathcal{I}^0} \\
S_{\mathcal{I}^0,1} & S_{\mathcal{I}^0,\mathcal{A}} & S_{\mathcal{I}^0,\mathcal{I}^+} & -S_{\mathcal{I}^0,\mathcal{I}^-} & I_{\mathcal{I}^0,\mathcal{I}^0}
\end{pmatrix}}_{=N}
\begin{pmatrix}
v_{\omega,1} \\
v_{\omega,\mathcal{A}} \\
v_{\omega,\mathcal{I}^+} \\
-v_{\omega,\mathcal{I}^-} \\
v_{d,\mathcal{I}^0}
\end{pmatrix}
$$

$$
+
\begin{pmatrix}
0 \\
0 \\
v_{d,\mathcal{I}^+} \\
-v_{d,\mathcal{I}^-} \\
0
\end{pmatrix}
=
\underbrace{\begin{pmatrix}
E_1 - S_{1,\mathcal{I}^0}^T v_{\omega,\mathcal{I}^0} - c \\
E_{\mathcal{A}} - S_{\mathcal{A},\mathcal{I}^0}^T v_{\omega,\mathcal{I}^0} - v_{d,\mathcal{A}} \\
E_{\mathcal{I}^+} - S_{\mathcal{I}^+,\mathcal{I}^0}^T v_{\omega,\mathcal{I}^0} \\
-E_{\mathcal{I}^-} + S_{\mathcal{I}^-,\mathcal{I}^0}^T v_{\omega,\mathcal{I}^0} \\
E_{\mathcal{I}^0} - S_{\mathcal{I}^0,\mathcal{I}^0}^T v_{\omega,\mathcal{I}^0}
\end{pmatrix}}_{=P}. \tag{26}
$$

Since $\boldsymbol{X}_{\bar{\mathcal{A}}}$ is full-rank,

$$
\begin{pmatrix}
S_{1,1} & S_{1,\mathcal{A}} & S_{1,\mathcal{I}^+} & S_{1,\mathcal{I}^-} \\
S_{\mathcal{A},1} & S_{\mathcal{A},\mathcal{A}} & S_{\mathcal{A},\mathcal{I}^+} & S_{\mathcal{A},\mathcal{I}^-} \\
S_{\mathcal{I}^+,1} & S_{\mathcal{I}^+,\mathcal{A}} & S_{\mathcal{I}^+,\mathcal{I}^+} & S_{\mathcal{I}^+,\mathcal{I}^-} \\
S_{\mathcal{I}^-,1} & S_{\mathcal{I}^-,\mathcal{A}} & S_{\mathcal{I}^-,\mathcal{I}^+} & S_{\mathcal{I}^-,\mathcal{I}^-}
\end{pmatrix}
= (1/n)\left(X_{1\cup\mathcal{A}\cup\mathcal{I}^+\cup\mathcal{I}^-}^T X_{1\cup\mathcal{A}\cup\mathcal{I}^+\cup\mathcal{I}^-}\right) \tag{27}
$$

is nonsingular (in fact, positive definite). Denote the $4 \times 4$ principal block submatrix of $N$ by

$$
N_1 = \begin{pmatrix}
S_{1,1}+b & S_{1,\mathcal{A}} & S_{1,\mathcal{I}^+} & -S_{1,\mathcal{I}^-} \\
S_{\mathcal{A},1} & S_{\mathcal{A},\mathcal{A}} & S_{\mathcal{A},\mathcal{I}^+} & -S_{\mathcal{A},\mathcal{I}^-} \\
S_{\mathcal{I}^+,1} & S_{\mathcal{I}^+,\mathcal{A}} & S_{\mathcal{I}^+,\mathcal{I}^+} & -S_{\mathcal{I}^+,\mathcal{I}^-} \\
-S_{\mathcal{I}^-,1} & -S_{\mathcal{I}^-,\mathcal{A}} & -S_{\mathcal{I}^-,\mathcal{I}^+} & S_{\mathcal{I}^-,\mathcal{I}^-}
\end{pmatrix}
\in \mathbb{R}^{|\bar{\mathcal{A}}|\times|\bar{\mathcal{A}}|} \tag{28}
$$

so that $N = \begin{pmatrix} N_1 & 0 \\ C & I_{\mathcal{I}^0,\mathcal{I}^0} \end{pmatrix}$ with

$$
C = (S_{\mathcal{I}^0,1}, S_{\mathcal{I}^0,\mathcal{A}}, S_{\mathcal{I}^0,\mathcal{I}^+}, -S_{\mathcal{I}^0,\mathcal{I}^-}). \tag{29}
$$

Then $N_1$ is also nonsingular since $b > 0$, and thus

$$
N^{-1} = \begin{pmatrix} N_1^{-1} & 0 \\ -CN_1^{-1} & I_{\mathcal{I}^0,\mathcal{I}^0} \end{pmatrix}.
$$

Therefore,

$$
\boldsymbol{S}v_\omega + v_d = E \iff
\begin{pmatrix}
v_{\omega,1} \\
v_{\omega,\mathcal{A}} \\
v_{\omega,\mathcal{I}^+} \\
-v_{\omega,\mathcal{I}^-} \\
v_{d,\mathcal{I}^0}
\end{pmatrix}
= N^{-1}\left[ P -
\begin{pmatrix}
0 \\
0 \\
v_{d,\mathcal{I}^+} \\
-v_{d,\mathcal{I}^-} \\
0
\end{pmatrix}\right]. \tag{30}
$$

If we let the $2 \times 2$ principal diagonl block submatrix of $N_1$ be $N_2 = \begin{pmatrix} S_{1,1} + b & S_{1,\mathcal{A}} \\ S_{\mathcal{A},1} & S_{\mathcal{A},\mathcal{A}} \end{pmatrix}$, then the

bottom right $2 \times 2$ block of $N_1^{-1}$ is the Schur complement of $N_2$ in $N_1$:

$$M := \left[ \begin{pmatrix} S_{\mathcal{I}^+,\mathcal{I}^+} & -S_{\mathcal{I}^+,\mathcal{I}^-} \\ -S_{\mathcal{I}^-,\mathcal{I}^+} & S_{\mathcal{I}^-,\mathcal{I}^-} \end{pmatrix} - \begin{pmatrix} S_{\mathcal{I}^+,1} & S_{\mathcal{I}^+,\mathcal{A}} \\ -S_{\mathcal{I}^-,1} & -S_{\mathcal{I}^-,\mathcal{A}} \end{pmatrix} N_2^{-1} \begin{pmatrix} S_{1,\mathcal{I}^+} & -S_{1,\mathcal{I}^-} \\ S_{\mathcal{A},\mathcal{I}^+} & -S_{\mathcal{A},\mathcal{I}^-} \end{pmatrix} \right]^{-1} \tag{31}$$

Thus,

$$\boldsymbol{S}v_\omega + v_d = E \implies \begin{pmatrix} v_{\omega,\mathcal{I}^+} \\ -v_{\omega,\mathcal{I}^-} \end{pmatrix} = M \begin{pmatrix} -v_{d,\mathcal{I}^+} \\ v_{d,\mathcal{I}^-} \end{pmatrix} + [N_1^{-1}]_{3:4} P, \tag{32}$$

where $[N_1^{-1}]_{3:4}$ refers to the submatrix of $N_1^{-1}$ spanning the third and fourth row blocks.

From the property of the Schur complement, $M$ is positive definite if and only if $N_1$ is positive definite. To see this, write

$$N_1 = I_0 \begin{pmatrix} S_{1,1} + b & S_{1,\mathcal{A}} & S_{1,\mathcal{I}^+} & S_{1,\mathcal{I}^-} \\ S_{\mathcal{A},1} & S_{\mathcal{A},\mathcal{A}} & S_{\mathcal{A},\mathcal{I}^+} & S_{\mathcal{A},\mathcal{I}^-} \\ S_{\mathcal{I}^+,1} & S_{\mathcal{I}^+,\mathcal{A}} & S_{\mathcal{I}^+,\mathcal{I}^+} & S_{\mathcal{I}^+,\mathcal{I}^-} \\ S_{\mathcal{I}^-,1} & S_{\mathcal{I}^-,\mathcal{A}} & S_{\mathcal{I}^-,\mathcal{I}^+} & S_{\mathcal{I}^-,\mathcal{I}^-} \end{pmatrix} I_0,$$

where

$$I_0 = \begin{pmatrix} I_{1,1} & & & \\ & I_{\mathcal{A},\mathcal{A}} & & \\ & & I_{\mathcal{I}^+,\mathcal{I}^+} & \\ & & & -I_{\mathcal{I}^-,\mathcal{I}^-} \end{pmatrix}$$

is symmetric and orthogonal. The middle matrix is (27) that is positive definite.

If we let

$$\mathbf{x} = \begin{pmatrix} -v_{d,\mathcal{I}^+} + \omega_{\mathcal{I}^+} \\ v_{d,\mathcal{I}^-} - \omega_{\mathcal{I}^-} \end{pmatrix}, \quad \mathbf{y} = \begin{pmatrix} v_{\omega,\mathcal{I}^+} + \omega_{\mathcal{I}^+} \\ -v_{\omega,\mathcal{I}^-} - \omega_{\mathcal{I}^-} \end{pmatrix}, \tag{33}$$

then we see

$$\langle \mathbf{x}, \mathbf{y} \rangle = 0 \quad \text{and} \quad \mathbf{y} = M\mathbf{x} + q,$$

where

$$q = -M \begin{pmatrix} \omega_{\mathcal{I}^+} \\ -\omega_{\mathcal{I}^-} \end{pmatrix} + [N_1^{-1}]_{3:4} P + \begin{pmatrix} \omega_{\mathcal{I}^+} \\ -\omega_{\mathcal{I}^-} \end{pmatrix}. \tag{34}$$

from (23) and (32). In other words, the $\mathbf{x}$ and $\mathbf{y}$ in (33) jointly solve the LCP (17) with the $M$ and $q$ given in (31) and (34). In fact, (17) has a unique solution, since a finite-dimensional LCP of the form (17) has a unique solution if $M$ is a $P$-matrix, meaning all of its principal minors are positive. A symmetric positive definite matrix is a $P$-matrix.

For the "if" part, suppose $(\mathbf{x}, \mathbf{y})$ solves the LCP (17) with the $M$ and $q$ given in (31) and (34). Then $(\mathbf{x}, \mathbf{y})$ is unique. Let

$$\begin{pmatrix} v_{d,\mathcal{I}^+} \\ v_{d,\mathcal{I}^-} \end{pmatrix} = \begin{pmatrix} -\mathbf{x}_{\mathcal{I}^+} \\ \mathbf{x}_{\mathcal{I}^-} \end{pmatrix} + \begin{pmatrix} \omega_{\mathcal{I}^+} \\ \omega_{\mathcal{I}^-} \end{pmatrix} \quad \text{for} \quad \mathbf{x} = \begin{pmatrix} \mathbf{x}_{\mathcal{I}^+} \\ \mathbf{x}_{\mathcal{I}^-} \end{pmatrix} \tag{35}$$

and

$$\begin{pmatrix} v_{\omega,\mathcal{I}^+} \\ -v_{\omega,\mathcal{I}^-} \end{pmatrix} = \left[ N^{-1} \left( P - \begin{pmatrix} 0 \\ 0 \\ v_{d,\mathcal{I}^+} \\ -v_{d,\mathcal{I}^-} \\ 0 \end{pmatrix} \right) \right]_{3:4}.$$

Then

$$\begin{pmatrix} v_{\omega,\mathcal{I}^+} \\ -v_{\omega,\mathcal{I}^-} \end{pmatrix} = \left[ N^{-1} \left( P - \begin{pmatrix} 0 \\ 0 \\ -\mathbf{x}_{\mathcal{I}^+} + \omega_{\mathcal{I}^+} \\ -\mathbf{x}_{\mathcal{I}^-} - \omega_{\mathcal{I}^-} \\ 0 \end{pmatrix} \right) \right]_{3:4}$$

$$= [N_1^{-1}]_{3:4} P - M \begin{pmatrix} \omega_{\mathcal{I}^+} \\ -\omega_{\mathcal{I}^-} \end{pmatrix} + M\mathbf{x} = M\mathbf{x} + q - \begin{pmatrix} \omega_{\mathcal{I}^+} \\ -\omega_{\mathcal{I}^-} \end{pmatrix} = \mathbf{y} - \begin{pmatrix} \omega_{\mathcal{I}^+} \\ -\omega_{\mathcal{I}^-} \end{pmatrix},$$

which implies (14)–(16). □

**Proof of Theorem 3.4**: From the uniqueness of the solution to LCP (17), the $v^k$ in (11) is well-defined: We know

$$v^k_{\omega,\mathcal{I}^0} = -\omega^k_{\mathcal{I}^0}, \quad v^k_{d,\mathcal{A}} = \lambda(\mathbf{1}^\top_{\mathcal{A}^+}, -\mathbf{1}^\top_{\mathcal{A}^-})^\top - d^k_{\mathcal{A}}$$

from (8) and (12). That

$$\begin{pmatrix} v^k_{d,\mathcal{I}^+} \\ v^k_{d,\mathcal{I}^-} \end{pmatrix} = \begin{pmatrix} -\mathbf{x}_{\mathcal{I}^+} \\ \mathbf{x}_{\mathcal{I}^-} \end{pmatrix} + \begin{pmatrix} \omega^k_{\mathcal{I}^+} \\ \omega^k_{\mathcal{I}^-} \end{pmatrix}$$

follows from (35); the $M$ and $q$ for solving $(\mathbf{x}, \mathbf{y})$ in LCP (17) are given in (31) and (34). Also, from (24),

$$v^k_{d,1} = b^k v^k_{\omega,1} - c^k,$$

where $b^k$ and $c^k$ can be deduced from (25). Finally, with the above quantities computed, we have from (30) and (26),

$$\begin{pmatrix} v^k_{\omega,1} \\ v^k_{\omega,\mathcal{A}} \\ v^k_{\omega,\mathcal{I}^+} \\ -v^k_{\omega,\mathcal{I}^-} \\ v^k_{d,\mathcal{I}^0} \end{pmatrix} = N^{-1}\left[ P^k - \begin{pmatrix} 0 \\ 0 \\ v^k_{d,\mathcal{I}^+} \\ -v^k_{d,\mathcal{I}^-} \\ 0 \end{pmatrix} \right]$$

with

$$N^{-1} = \begin{pmatrix} N_1^{-1} & 0 \\ -CN_1^{-1} & I \end{pmatrix} \quad \text{and} \quad P^k = \begin{pmatrix} E^k_1 - S^T_{1,\mathcal{I}^0} v^k_{\omega,\mathcal{I}^0} - c^k \\ E^k_{\mathcal{A}} - S^T_{\mathcal{A},\mathcal{I}^0} v^k_{\omega,\mathcal{I}^0} - v^k_{d,\mathcal{A}} \\ E^k_{\mathcal{I}^+} - S^T_{\mathcal{I}^+,\mathcal{I}^0} v^k_{\omega,\mathcal{I}^0} \\ -E^k_{\mathcal{I}^-} + S^T_{\mathcal{I}^-,\mathcal{I}^0} v^k_{\omega,\mathcal{I}^0} \\ E^k_{\mathcal{I}^0} - S^T_{\mathcal{I}^0,\mathcal{I}^0} v^k_{\omega,\mathcal{I}^0} \end{pmatrix},$$

where $N_1$ and $C$ are given in (28) and (29) and $E^k = -\mathbf{S}\omega^k - d^k$. □

## F    Proofs of Theorem 3.8

### F.1    Proof of Theroem 3.8

To verify the global convergence of Algorithm 1, we utilize some regularity conditions of $F$ as in [28, 25]. Consider following lemmas:

**Lemma F.1.** *The proximal gradient step in Algorithm 1 occurs only a finite number of times.*

**Lemma F.2.** *There exists a constant $L > 0$ and a neighborhood $\mathcal{N}$ of $x^*$ such that for all $x \in \mathcal{N}$,*

$$\|F'(x; v)\| \geq L\|v\| \tag{36}$$

*holds for all vectors $v \in \mathbb{R}^{2p}$.*

**Lemma F.3.** *For every sequence $\{x^k\}$ converging to $x^\dagger$, $\{v^k\}$ converging to $v^\dagger$, and positive scalar sequence $\{t_k\}$ converging to zero,*

$$\limsup_{k \to \infty} \frac{\theta(x^k + t_k v^k) - \theta(x^k)}{t_k} \leq \lim_{k \to \infty} F(x^k)^T F'(x^k; v^k)$$

**Proof of Theorem 3.8**: We exclude the trivial case where $x^k$ exactly hits the optimal solution during the iteration, in other words, $\theta(x^k) > 0$. First, we show that $m_k$ in (18) always exists for all Newton iterations. To see this assume that

$$\theta(x^k + \rho^m v^k) - \theta(x^k) > -2\sigma\rho^m \theta(x^k), \tag{37}$$

holds for all $m \in \{m_j\}$ where $m_j$ is an infinite increasing integer sequence. Then, dividing both sides by $\rho^{m_j}$ and taking the limit as $j \to \infty$, we have

$$-2\sigma\theta(x^k) \leq \lim_{j \to \infty} \frac{\theta(x^k + \rho^{m_j} v^k) - \theta(x^k)}{\rho^{m_j}} = F'(x^k; v^k)^T F(x^k) = -2\theta(x^k).$$

as we assumed that $\sigma \in (0, 1)$ and $\theta(x^k) > 0$, we have a contradiction. So, there always exists $m_k$ such that (18) holds. Since $\{\theta(x^k)\}$ is a strictly decreasing sequence bounded above zero, its limit exists and therefore,

$$0 = \lim_{k\to\infty} \left\{\theta(x^k) - \theta(x^{k+1})\right\} \geq 2\sigma \limsup_{k\to\infty}(\rho^{m_k}\theta(x^k)) = 0.$$

If $\limsup_{k\to\infty} \rho^{m_k} > 0$, then $\lim_{k\to\infty} \theta(x^k) = 0$. Since we assume that $\mathcal{I}^{+^*} \cup \mathcal{I}^{-^*} = \emptyset$ and $X_{\bar{A}^*}$ has full column rank, every (in particular, the unique) generalized jacobian $\partial F(x^*)$ is nonsingular. Then, by Clarke's inverse function theorem [3], $F$ admits a locally Lipschitz continuous inverse in a neighborhood of $x^*$. Consequently, for all sufficiently large $k$, we have $x^k = F^{-1}(F(x^k)) \to F^{-1}(0) = x^*$.

If $\limsup_{k\to\infty} \rho^{m_k} = 0$, then we have $\rho^{m_k}\theta(x^k) \to 0$ and

$$\theta(x^k + \rho^{m_k-1}v^k) - \theta(x^k) > -2\sigma\rho^{m_k-1}\theta(x^k).$$

Consider a subsequence of $\{x^k : k \in K\}$ that converges to $x^\dagger$ such that $\bar{\mathcal{A}}^\dagger = \bar{\mathcal{A}}^*$. For all $k \in K$ large enough, $x^k \in \mathcal{N}$, where $\mathcal{N}$ is the neighborhood of $x^*$ specified in Lemma F.2. By Lemma F.2, it follows that $\|v^k\| \leq \frac{1}{L}\|F(x^k)\|$, which implies that the corresponding subsequence $\{v^k\}_{k\in K}$ is bounded. Thus, we can find a subsequence $k_1, k_2$ such that $x^{k_l}$ and $v^{k_l}$ converges into some $x^\dagger$ and $v^\dagger$, respectively. Then, by Lemma F.3, we have

$$\lim_{l\to\infty} F(x^{k_l})^T F'(x^{k_l}; v^{k_l}) \geq \lim_{l\to\infty} \frac{\theta(x^{k_l} + \rho^{m_k-1}v^{k_l}) - \theta(x^{k_l})}{\rho^{m_{k_l}-1}} \geq -2\sigma\theta(x^\dagger)$$

On the other hand, since $v^{k_l}$ is a Newton direction for (11),

$$\lim_{l\to\infty} F(x^{k_l})^T F'(x^{k_l}; v^{k_l}) = \lim_{l\to\infty} -2\theta(x^{k_l}) = -2\theta(x^\dagger).$$

Therefore, $\theta(x^\dagger) = 0$ and since the solution is unique, the accumulation point $x^\dagger$ coincides with $x^*$.

Finally, we establish the convergence of the full sequence $\{x^k\}$. Since $x^*$ is the unique solution to $F(x) = 0$, there exists $\delta > 0$ such that $x^*$ is the unique minimizer of $\|F(x)\|$ for $\|x - x^*\| \leq \delta$. Recall that for all $k$ large enough, $\|v^k\| \leq \frac{1}{L}\|F(x^k)\|$ from Lemma F.2. Since $\|F(x^k)\| \to 0$ as $k \to \infty$, it follows that $\|v^k\| \to 0$. Therefore, there exists a constant $\delta' \in (0, \delta)$ such that, we have

$$\|x^{k+1} - x^k\| \leq \|v^k\| \leq \delta - \delta'.$$

whenever $\|x^k - x^*\| \leq \delta'$. Let $\mu > 0$ denote the minimum value of $\|F(x)\|$ over the region $\delta' \leq \|x - x^*\| \leq \delta$. Suppose that $\|x^{k_0} - x^*\| \leq \delta'$ and $\|F(x^{k_0})\| < \mu$. Then, since the sequence $\|F(x^k)\|$ is strictly decreasing, it follows that $\|F(x^{k_0+1})\| < \mu$. But, since $\|x^{k_0+1} - x^{k_0}\| \leq \delta - \delta'$, we also have that

$$\|x^{k_0+1} - x^*\| \leq \|x^{k_0+1} - x^{k_0}\| + \|x^{k_0} - x^*\| \leq \delta.$$

Hence, by the definition of $\mu$, we have $\|x^{k_0+1} - x^*\| \leq \delta'$. By induction, it follows that $\|x^k - x^*\| \leq \delta'$ and $\|F(x^k)\| < \mu$ for all $k \geq k_0$. Since $x^*$ is the unique accumulation point of $\{x^k\}$ within this neighborhood, we conclude that the entire sequence $\{x^k\}$ converges to $x^*$.

We now characterize the local convergence behavior of the iterates. By [30, Theorem 4.3], since $x^k \to x^*$, the unique solution to $F(x) = 0$, and $F'(\cdot; \cdot)$ in (12) is semicontinuous of degree 2 [30, Lemma 2.3], the sequence $\{x^k\}$ converges quadratically to $x^*$ in its neighborhood and $\rho^m$ eventually becomes 1. $\qquad\square$

## F.2 Proof of Lemma F.1

The proximal gradient algorithm of [19] generates an iterate sequence to converge linearly to the unique solution $x^*$. We distinguish between two contexts in which the proximal gradient step is invoked: the inner loop (lines 14–16) and the outer loop (lines 12–17).

First consider the inner loop. Let $\{x_{\text{prox}}^{k_j}\}$ denote the sequence of iterates computed by the proximal gradient update. This update is applied iteratively from the initial point $x_{\text{prox}}^{k_0} = x^k$. Assume $\theta(x_{\text{prox}}^{k_0}) > 0$. (Otherwise we are done.) Since $x_{\text{prox}}^{k_j} \to x^*$ as $j \to \infty$, $\theta(x_{\text{prox}}^{k_j}) \to 0$ as $j \to \infty$, and

by linear convergence, there exists a finite index $J$ such that $\theta(x_{\text{prox}}^{k_J}) \leq \theta(x_{\text{prox}}^{k_0})$. Thus, the inner loop terminates in finitely many iterations.

Next, consider the proximal gradient phase of the outer loop. This phase is potentially invoked until the iterate enters a neighborhood $\mathcal{N}$ of $x^* = (\omega^*, d^*)$ where $\bar{\mathcal{A}} = \bar{\mathcal{A}}^*$ holds (such a neighborhood exists due to the assumption of Theorem 3.8). If the iterate does not enter $\mathcal{N}$ after a finite number of invocations, then $\theta(x_{\text{prox}}^{k_j}) \nrightarrow 0$, a contradiction.

Once the (outer) sequence $\{x^k\}$ enters $\mathcal{N}$, within this neighborhood the rank condition for local B-semismooth Newton convergence is satisfied, so the algorithm remains in the Newton phase. Consequently, the proximal gradient step is invoked only a finite number of times throughout the entire procedure. $\qquad\square$

### F.3 Proof of Lemma F.2

Let
$$
\mathcal{A} = \{i \in \{2, \ldots, p\} : |\omega_i + d_i| > \lambda\},
$$
$$
\mathcal{I} = \{i \in \{2, \cdots, p\} : |\omega_i + d_i| \leq \lambda\}.
$$

Recall that the Bouligand derivative $F'(\omega, d; v)$ can be expressed with the generalized Jacobian $G_{\mathcal{A}}(\omega, d)$ as

$$
F'(\omega, d; v) = G_{\mathcal{A}}(\omega, d)v = \begin{pmatrix} X_1^T X_1 & X_1^T X_{\mathcal{A} \cup \mathcal{I}} & 1 & 0 \\ X_{\mathcal{A} \cup \mathcal{I}}^T X_1 & X_{\mathcal{A} \cup \mathcal{I}}^T X_{\mathcal{A} \cup \mathcal{I}} & 0 & \boldsymbol{I} \\ z + \frac{1}{2} & 0 & z - \frac{1}{2} & 0 \\ 0 & I_{\mathcal{I}} & 0 & -I_{\mathcal{A}} \end{pmatrix} v
$$

for the differentiable points $(|\omega_i + d_i| \neq \lambda)$ where $z = -\frac{\omega_1 + d_1}{2\sqrt{(\omega_1 + d_1)^2 + 4}}$. Since we know that the value of the $(p + i)$-th element of $F'(\omega, d; v)$ for $i \in \mathcal{I}^{\pm}$ is $-v_{d,i}$ or $v_{\omega,i}$, we can construct the following inequality:
$$
\|F'(\omega, d; v)\|^2 \geq \inf_{\mathcal{B} \subseteq \mathcal{I}^{\pm}} \|G_{\mathcal{A} \cup \mathcal{B}}(\omega, d)v\|
$$

Thus, it suffices to show that the generalized Jacobian $G_{\mathcal{A}}(\omega, d)$ is nonsingular. Then, it can be deduced that
$$
\|G_{\mathcal{A}}^{-1}(\omega, d)\|\|G_{\mathcal{A}}(\omega, d)v\| \geq \|v\|,
$$

and thus, $L$ in (36) holds for the minimum value of $\|G_{\mathcal{A}}^{-1}(\omega, d)\|$. Since $F$ is continuous and $\mathcal{I}^{+^*} \cup \mathcal{I}^{-^*} = \emptyset$, there exists a neighborhood $\mathcal{N}$ of $(\omega^*, d^*)$ such that for all $(\omega, d) \in \mathcal{N}$, we have $\bar{\mathcal{A}} = \bar{\mathcal{A}}^*$. Since $\boldsymbol{X}_{\bar{\mathcal{A}}^*}$ has full column rank, $\boldsymbol{X}_{\bar{\mathcal{A}}}$ also has full column rank for all $(\omega, d)$ sufficiently close to $(\omega^*, d^*)$. We already proved that such $G_{\mathcal{A}}$ has an inverse in Appendix C. $\qquad\square$

### F.4 Proof of Lemma F.3

Note that
$$
\lim_{k \to \infty} \frac{F_i^2(x^k + t_k v^k) - F_i^2(x^k)}{2t_k} \leq \lim_{k \to \infty} F(x^k)^T F'(x^k; v^k) \tag{38}
$$

holds as the equality within its domain of $x$ for $i = 1, \cdots, p + 1$, and the equality also holds for $i = p + 1, \cdots, 2p$ where $|\omega_j + d_j| \neq \lambda$ $(j = 2, \cdots, p)$, as $F_i$'s are differentiable and $F_i'$'s are continuous at these points. Hence, it suffices to prove that (38) also holds for non-differential points $|\omega + d| = \lambda$. Suppose that $(\omega^k, d^k)$ converges to $(\omega^{\dagger}, d^{\dagger})$ and $(v_{\omega}^k, v_d^k)$ converges to $(v_{\omega}^{\dagger}, v_d^{\dagger})$. Without loss of generality, let $\omega^{\dagger} + d^{\dagger} = \lambda$. Let $k_1, k_2, \cdots$ be any increasing sequence of a positive integer. Then, for any $i \in \{p + 1, \cdots, 2p\}$, we can find some subsequence $k_{m_l}$ where

$$
\lim_{l \to \infty} \frac{F_i((\omega^{k_{m_l}} + t_{k_{m_l}} v_{\omega}^{k_{m_l}}, d^{k_{m_l}} + t_{k_{m_l}} v_d^{k_{m_l}})) - F_i((\omega^{k_{m_l}}, d^{k_{m_l}}))}{t_{k_{m_l}}} \tag{39}
$$

exists and falls into at least one of the following cases:

1. $\omega^{k_{m_l}} + d^{k_{m_l}} > \lambda$ and $\omega^{k_{m_l}} + d^{k_{m_l}} + t_{k_{m_l}}(v_\omega^{k_{m_l}} + v_d^{k_{m_l}}) \geq \lambda$. Then,

$$(39) = \lim_{l \to \infty} \frac{t_{k_{m_l}} v_\omega^{k_{m_l}} - t_{k_{m_l}}(v_\omega^{k_{m_l}} + v_d^{k_{m_l}})}{t_{k_{m_l}}} = -v_d^\dagger = \lim_{l \to \infty} F_i'(x^{k_{m_l}}; v^{k_{m_l}}).$$

2. $\omega^{k_{m_l}} + d^{k_{m_l}} < \lambda$ and $\omega^{k_{m_l}} + d^{k_{m_l}} + t_{k_{m_l}}(v_\omega^{k_{m_l}} + v_d^{k_{m_l}}) \geq \lambda$. Then,

$$(39) = \lim_{l \to \infty} \frac{t_{k_{m_l}} v_\omega^{k_{m_l}} - (\omega^{k_{m_l}} + d^{k_{m_l}} + t_{k_{m_l}}(v_\omega^{k_{m_l}} + v_d^{k_{m_l}}) - \lambda)}{t_{k_{m_l}}}$$

$$\leq \lim_{l \to \infty} (t_{k_{m_l}} v_\omega^{k_{m_l}})/t_{k_{m_l}} = v_\omega^\dagger = \lim_{l \to \infty} F_i'(x^{k_{m_l}}; v^{k_{m_l}}).$$

3. $\omega^{k_{m_l}} + d^{k_{m_l}} > \lambda$ and $\omega^{k_{m_l}} + d^{k_{m_l}} + t_{k_{m_l}}(v_\omega^{k_{m_l}} + v_d^{k_{m_l}}) \leq \lambda$. Then,

$$(39) = \lim_{l \to \infty} \frac{t_{k_{m_l}} v_\omega^{k_{m_l}} + (\omega^{k_{m_l}} + d^{k_{m_l}} - \lambda)}{t_{k_{m_l}}}$$

$$\leq \lim_{l \to \infty} \frac{t_{k_{m_l}} v_\omega^{k_{m_l}} - t_{k_{m_l}}(v_\omega^{k_{m_l}} + v_d^{k_{m_l}})}{t_{k_{m_l}}} = -v_d^\dagger = \lim_{l \to \infty} F_i'(x^{k_{m_l}}; v^{k_{m_l}}).$$

4. $\omega^{k_{m_l}} + d^{k_{m_l}} < \lambda$ and $\omega^{k_{m_l}} + d^{k_{m_l}} + t_{k_{m_l}}(v_\omega^{k_{m_l}} + v_d^{k_{m_l}}) \leq \lambda$. Then,

$$(39) = \lim_{l \to \infty} (t_{k_{m_l}} v_\omega^{k_{m_l}})/t_{k_{m_l}} = v_\omega^\dagger = \lim_{l \to \infty} F_i'(x^{k_{m_l}}; v^{k_{m_l}}).$$

5. $\omega^{k_{m_l}} + d^{k_{m_l}} = \lambda$. Then,

$$(39) = \lim_{l \to \infty} (t_{k_{m_l}} v_\omega^{k_{m_l}} - t_{k_{m_l}} \max\{v_\omega^{k_{m_l}} + v_d^{k_{m_l}}, 0\})/t_{k_{m_l}}$$

$$= \min\{v_\omega^\dagger, -v_d^\dagger\} \leq \lim_{l \to \infty} F_i'(x^{k_{m_l}}; v^{k_{m_l}}).$$

Hence, we can conclude that

$$\limsup_{k \to \infty} \frac{F_i(x^k + t_k v^k) - F_i(x^k)}{t_k} \leq \lim_{k \to \infty} F_i'(x^k; v^k)$$

holds for all $i$. Then, using the continuity of $F$ it follows that

$$\limsup_{k \to \infty} \frac{\theta(x^k + t_k v^k) - \theta(x^k)}{t_k} \leq \lim_{k \to \infty} F(x^k)^T F'(x^k; v^k)$$

with some simple arithmetic. $\qquad \square$

## G    Additional Experiments

### G.1    Ablation study: roles of the proximal update and LCP solving

Algorithm 1 includes two components other than the semismooth Newton steps, that are the proximal update steps and the LCP solve step. To disentangle the roles of these two components, we replicated the setup of Section 4.1 and ran each configuration for 50 iterations under three ablations: (i) disabling only the proximal update steps, (ii) disabling only the LCP solve step, and (iii) disabling both. The LCP solve step is invoked only when, for a given row, $\mathcal{I}^+ \cup \mathcal{I}^- \neq \emptyset$. In this ablation study, "disabling the LCP solve step" means reclassifying all indices in $\mathcal{I}^+ \cup \mathcal{I}^-$ as inactive by assigning them to $\mathcal{I}^0$, thereby bypassing the LCP solve subroutine. Recall that with both components enabled, Algorithm 1 terminated within 30 iterations.

At $\lambda = 0.10$, disabling the proximal steps prevented the algorithm from converging within 50 iterations, regardless of whether the LCP solve component was enabled. By contrast, convergence was achieved whenever the proximal gradient component was retained, even without the LCP solve

comeponent. In the original run at this value of $\lambda$, the proximal update was invoked in 229 out of 1000 rows in the first iteration, indicating its importance in the denser regime. For $\lambda = 0.15, 0.2$, the algorithm converged under all ablation settings. In the original runs, at most one proximal update (out of 1000 rows) occurred in the first iteration, after which the semismooth Newton updates predominated. Thus, while the proximal gradient steps can be dispensable in these sparser regimes, it remains a practical safeguard unless the sparsity level is known a priori.

The LCP solve step was activated only on the kink set $\{i : |\omega_i + d_i| = \lambda\}$. Since exact equality is numerically rare, such activations almost never occurred. Consistent with this observation, Table 4 shows that Algorithm 1 is largely insensitive to this event, and Table 5 reports that the LCP solve occurs for a negligible fraction of rows. Even with 10,000 Newton row updates per iteration, the LCP solve occurred in about one or fewer row on average, as reflected by the small average and maximum per-iteration fractions. Thus, the practical impact of the LCP module is limited, although it remains theoretically indispensable for correctness and robustness, especially on unseen data.

| $\lambda$ | Full method | No Proximal | No LCP | No Proximal + No LCP |
|---|---|---|---|---|
| 0.10 | Converged (30) | Diverged | Converged (30) | Diverged |
| 0.15 | Converged (22) | Converged (26) | Converged (22) | Converged (26) |
| 0.20 | Converged (14) | Converged (14) | Converged (14) | Converged (14) |

Table 4: Convergence within 50 iterations under ablations of the proximal and LCP modules.

| $\lambda$ | Avg. frac. LCP / iter. | Max frac. LCP / iter. | Last iter. LCP occurred | Total iters. |
|---|---|---|---|---|
| 0.20 | 0.00005 | 0.0011 | 62 | 159 |
| 0.45 | 0.00004 | 0.0012 | 29 | 69 |
| 0.60 | 0.00012 | 0.0045 | 31 | 63 |

Table 5: Incidence of LCP updates in the original runs. "Frac." denotes the fraction of rows updated by LCP in an iteration.

## G.2 Sensitivity to the choice of regularization parameter $\lambda$

We assessed the convergence behavior of Algorithm 1 across a range of regularization parameters under the same experimental setup as described in Section 4.3, focusing on real-world datasets. The baseline choice $\lambda = 0.450$ was selected by the extended Bayesian information criterion in [19]. To examine sensitivity, we varied $\lambda$ from 0.325 to 0.575 with an increment of 0.025 and recorded the iteration count, the number of nonzeros in the estimated precision matrix, and the resulting fraction of nonzero elements (density). The result is summarized in Table 6.

The change of the density remained modest within the range $\lambda \in [0.4, 0.5]$, indicating that the sparsity pattern is relatively insensitive around the "optimal" $\lambda = 0.450$. Across the tested $\lambda$ values, Algorithm 1 converged within a few tens of iterations; while the iteration count tended to increase mildly as $\lambda$ decreases (i.e., as the solution becomes denser), the overall convergence behavior remained stable.

## G.3 Convergence across densities and alternate random graph topologies

In this section, we further investigate the trade-off between graph density and convergence using alternative random graph structures. In Section 4.2, the ground truth precision matrix $\Theta^*$ was constructed based on an Erdős–Rényi graph, where edges are sampled uniformly at random. To capture a broader variety of graph structures, we additionally consider two alternative random graph topologies: hub networks and scale-free graphs.

A hub network was generated using the following procedure: (1) for each cluster, $(300 \times \text{density})$ hub nodes were selected, each of which was connected to 66 randomly chosen nodes; (2) the remaining $(100 - 300 \times \text{density})$ nodes within each cluster were connected via an Erdős–Rényi graph, using the remaining intra-cluster edges.

| $\lambda$ | Iterations | Nonzeros | Density (%) |
|---|---|---|---|
| 0.325 | 103 | 5,494,644 | $5.89 \times 10^{-3}$ |
| 0.350 | 69 | 4,806,952 | $5.15 \times 10^{-3}$ |
| 0.375 | 78 | 4,214,966 | $4.52 \times 10^{-3}$ |
| 0.400 | 67 | 3,699,259 | $3.96 \times 10^{-3}$ |
| 0.425 | 94 | 3,248,215 | $3.48 \times 10^{-3}$ |
| 0.450 | 64 | 2,852,968 | $3.06 \times 10^{-3}$ |
| 0.475 | 67 | 2,505,706 | $2.68 \times 10^{-3}$ |
| 0.500 | 71 | 2,200,810 | $2.36 \times 10^{-3}$ |
| 0.525 | 56 | 1,930,194 | $2.07 \times 10^{-3}$ |
| 0.550 | 66 | 1,692,240 | $1.81 \times 10^{-3}$ |
| 0.575 | 65 | 1,481,495 | $1.59 \times 10^{-3}$ |

Table 6: Iteration count, number of nonzero elements, and the fraction of nonzero elements in the estimated precision matrix for the LIHC dataset with varying regularization paremeter $\lambda$.

A scale-free graph was generated so that the degree $k$ of the nodes has a probability $P(k) \sim k^{-2.3}$.

Each graph comprised 10 clusters with 100 nodes per cluster. The total number of edges was determined by the graph density defined in Section 4.2, with $80\%$ of the edges allocated to intra-cluster connections and the remaining $20\%$ to inter-cluster connections. The resulting graphs are summarized in Table 7.

| Density | Hub Nodes | Intra-cluster Edges per Cluster |
|---|---|---|
| 0.03 | 9 | 1198 |
| 0.05 | 15 | 1998 |
| 0.10 | 30 | 3996 |

Table 7: The number of hub nodes and intra-cluster edges per cluster according to graph density

Following the experimental setup described in Section 4.2, the regularization parameter $\lambda$ was chosen so that the estimated precision matrix exhibits a sparsity level similar to that of the true matrix $\Theta^*$. We then assessed whether Algorithm 1 converges within 300 iterations across various graph types and density levels $(0.03, 0.05, \text{and } 0.1)$. The results are summarized in Tables 8–11.

We observe that the convergence behavior in different graph density levels is primarily determined by the overall density of the true precision matrix (i.e., graph structure), rather than by the specific graph structures. Intuitively, although the graph structures considered in this experiment may exhibit diverse local structural patterns, their overall density was controlled to remain constant across all settings. Since the applicability of the B-semismooth Newton method depends more on overall graph density than on local structural variations, the observed results are consistent with this interpretation.

**Hub Network**

| Density $\setminus \lambda$ | 0.2 | 0.15 | 0.1 | 0.09 | 0.08 | 0.07 |
|---|---|---|---|---|---|---|
| 0.03 | Y | Y | - | - | - | - |
| 0.05 | - | Y | Y | - | - | - |
| 0.1 | - | - | Y | Y | Y | N |

| Density $\setminus \lambda$ | 0.07 | 0.06 | 0.05 |
|---|---|---|---|
| 0.1 | 3% | 83.3% | 100% |

Table 8: Convergence within 300 iterations for varying graph density and $\lambda$

Table 9: Proportion of rows updated by proximal method after 300 iterations

**Scale-free graph**

| Density \ $\lambda$ | 0.2 | 0.15 | 0.1 | 0.09 | 0.08 | 0.07 |
|---|---|---|---|---|---|---|
| 0.03 | Y | Y | - | - | - | - |
| 0.05 | - | Y | Y | - | - | - |
| 0.1 | - | - | Y | Y | Y | N |

Table 10: Convergence within 300 iterations for varying graph density and $\lambda$

| Density \ $\lambda$ | 0.07 | 0.06 | 0.05 |
|---|---|---|---|
| 0.1 | 1.6% | 85.5% | 99.8% |

Table 11: Proportion of rows updated by proximal method after 300 iterations

