# OpenReview forum: "Partial Correlation Network Estimation by Semismooth Newton Methods"
_NeurIPS.cc/2025/Conference — NeurIPS 2025 poster_

### Official Review · Reviewer_jUps · 2025-06-24

**Clarity:** 4
**Significance:** 3
**Originality:** 3
**Rating:** 5
**Confidence:** 4

**Summary:**

The paper focuses on the problem of partial correlation network estimation. The classical approach to this is the graphical lasso problem.
Unfortunately, as the authors note, the graphical lasso struggles to scale to datasets where p>1000, owing to cubic scaling that results from the KKT condition. This is problematic, as in areas like genetics one routinely encounters problems where p>100000, for which graphical lasso would be prohibitively expensive.

Recently, a framework called ACCORD proposes solving an equivalent problem that is much easier to solve for large p. Existing approaches for solving this problem are based on the proximal gradient method, which while possessing scalable iterations, is slow to converge to an acceptable solution.
Indeed, a problem with p~300000, required more than 200 node hours of compute.

To address this the problem, the authors develop a bespoke semismooth-Newton method for solving the equivalent problem. Unlike the proximal gradient approach, the proposed method exhibits local quadratic convergence which ensures rapid convergence to a high accuracy solution once the iterates are close enough to the solution. In addition, global convergence from any initialization is ensured.
Despite being a Newton type method, the authors show that thanks to the structure of the problem, the resulting subproblems reduce to solving small linear systems or linear complementarity problems.
Numerical experiments validate the theory in the paper and demonstrates superior performance to the current state-of-art proximal gradient algorithm for the problem.

**Questions:**

- Can you clarify the relation of your use of the semismooth Newton method in relation to references [1,3] in the above?
This would solidify the novelty of the approach taken by the paper and would make connections with prior uses of semsimooth Newton for sparse learning problems.

**Ethical Concerns:**

["NO or VERY MINOR ethics concerns only"]

**Final Justification:**

The issues regarding the novelty of the method and contributions relative to prior methods utilizing Semismooth Newton for sparse learning was addressed. As I already thought the paper was worth accepting, I'll be maintaining my initial recommendation.

**Limitations:**

Yes

**Quality:**

4

**Strengths And Weaknesses:**

**Strengths**

- The presentation in the paper is excellent. The authors thoroughly explain the limitations of existing methodology for solving the partial correlation network estimation problem, and carefully explain each component of their algorithm and how it addresses the problem. Similarly, the proofs of the main results are presented clearly and with sufficient detail.

- The algorithm developed in the paper is efficient and comes with strong theoretical guarantees that address the limitations of existing algorithms.

- Numerical experiments show that the proposed algorithm outperforms the existing baseline on a real world large-scale genetics task, which shows its potential to yield better performance on real world tasks that are of interest to practitioners.

**Weaknesses**

- My main issue with the paper is that it misses some important related work. Semismooth Newton methods have been used for solving sparse learning problems for sometime, see for example [1-3].
Thus, the authors should cite these works and clarify how their approach differs from this prior work.

**Overall**

The paper makes a good contribution by developing a better solver for an important problem that frequently arises in large-scale genomics. I believe the method will be of interest to members of the community working in this area.
The main issue I would like to see the authors address is my question about how their method compares prior semismooth Newton methods for large-scale sparse learning.

**References**

[1] Li, X., Sun, D. and Toh, K.C., 2018. A highly efficient semismooth Newton augmented Lagrangian method for solving Lasso problems. SIAM Journal on Optimization, 28(1), pp.433-458.

[2] Lin, M., Liu, Y.J., Sun, D. and Toh, K.C., 2019. Efficient sparse semismooth Newton methods for the clustered Lasso problem. SIAM Journal on Optimization, 29(3), pp.2026-2052.

[3] Zhang, Y., Zhang, N., Sun, D. and Toh, K.C., 2020. An efficient Hessian based algorithm for solving large-scale sparse group Lasso problems. Mathematical Programming, 179, pp.223-263.

---

> ### Author Rebuttal · Authors · 2025-07-31
>
> ### Answer to Weakness 1 and Question 1
>
> The line of work you referred to, such as those solving Lasso, Clustered Lasso, and Sparse Group Lasso problems ([1]–[3]), falls into the semismooth Newton-based augmented Lagrangian method (SSNAL) framework. As the name suggests, this framework solves the problem via the inexact augmented Lagrangian method (ALM). ALM incurs an optimization subproblem, and SSNAL solves this subproblem via semismooth Newton. While the subproblem is solved at a locally superlinear (quadratic if it is strongly semismooth) rate, the convergence rate of the the outer ALM iteration is locally linear, even if the subproblem can be solved at a locally quadratic rate. Global convergence is also established.
>
> In contrast, our algorithm formulates the main problem (KKT equation) as a strongly semismooth equation and apply semismooth Newton. Globalization is achieved via damping instead of ALM. So our algorithmic design is fundamentally different from SSNAL: there is no subproblem to solve, and the convergence is locally quadratic. (If the KKT equation was merely semi smooth, then the local convergence is superlinear.)
>
> Another distinction is that SSNAL typically constructs augmented Lagrangian via the dual of the original (primal) problem, while we focus on the primal. It is worth pointing out that [2] applies the SSNAL framework directly to the primal in addition to the dual, but its design and analysis focus on low-dimensional settings where the number of observations significantly exceeds the number of features (i.e.,$n \gg p$) As such, it is not readily applicable to high-dimensional settings, which are the primary focus of our work.
>
> We appreciate the opportunity to clarify these differences and will explicitly cite and discuss the relevant SSNAL literature in the revised version to better position our contribution in the context of prior work.

---

> > ### Comment · Reviewer_jUps · 2025-08-05
> > **Reply to Rebuttal**
> >
> > I appreciate the authors clarifying their approach and contributions relative to past literature that uses Semismooth Newton methods for sparse learning. The response has addressed my concerns, and the discussion will make a valuable addition to the paper. I will maintain my current score.

---

### Official Review · Reviewer_7HYy · 2025-07-02

**Clarity:** 2
**Significance:** 3
**Originality:** 3
**Rating:** 5
**Confidence:** 2

**Summary:**

This paper introduces a semismooth Newton method for partial correlation network estimation, establishing a framework that ensures both local and global convergence while mitigating the high iteration costs of first-order methods. The method reformulates the KKT conditions as a nonlinear equation, employs Bouligand derivatives for global convergence, and addresses linear complementarity problems to optimize iteration direction. Experiments on synthetic and liver cancer genomic data demonstrate the method's quadratic convergence and computational efficiency, achieving a significant speedup over proximal gradient methods.

**Questions:**

Please see the detailed comments on Weaknesses and Questions.

**Ethical Concerns:**

["NO or VERY MINOR ethics concerns only"]

**Final Justification:**

Many thanks to the authors for the rebuttal. Most of my concerns were addressed. So I will raise my score to 5. Good luck.

**Limitations:**

No. Please see the detailed comments on Weaknesses and Questions.

**Quality:**

3

**Strengths And Weaknesses:**

Strengths:
1. The background analysis is comprehensive, focusing on computational bottlenecks in high-dimensional network estimation, such as matrix inversion complexity, with a clearly defined objective.
2. The innovative framework combines second-order information from semismooth Newton methods with B-derivative globalization, theoretically guaranteeing local quadratic convergence and global stability.
3. The experimental design is compelling, demonstrating effectiveness and scalability through comparisons with proximal gradient methods, tests across different graph densities, and applications to large-scale genomic data.

Weakness and Questions：
Major problems:
1. Lack of systematic analysis for hyperparameter selection and algorithm switching criteria: The paper does not clarify how regularization parameters (e.g., λ) or line search step sizes are chosen (e.g., via cross-validation). The switching conditions between the two-stage algorithm (local Newton vs. global B-derivative strategy), such as error thresholds or active set rank criteria, lack theoretical justification, potentially affecting convergence stability.
2. Incomplete ablation studies: Core components (e.g., semismooth Newton iteration, linear complementarity problem solving) are not ablated, precluding quantification of each module’s contribution to convergence speed (e.g., how removing B-derivative globalization affects convergence rate).
3. Unexplored applicability of quasi-Newton methods: While semismooth Newton methods avoid matrix diagonalization, the paper does not compare quasi-Newton methods (e.g., L-BFGS) for optimizing semismooth functions, missing a cross-validation of computational complexity and convergence properties.
4. Limited generalizability validation: Experiments are confined to partial correlation network estimation, without verifying the method’s applicability to unstructured optimization problems (e.g., general convex/non-convex functions). Dependencies on problem structures (e.g., row-wise separability) need further clarification.

Minor Problems:
Intuitive explanations for theoretical derivations (e.g., generalized Jacobian construction) are insufficient, potentially hindering reader comprehension.

---

> ### Author Rebuttal · Authors · 2025-07-31
>
> ## Answer to Major Problems
> ### Problem 1.
> We address each component of your concerns in turn:
>
> (1) Regularization parameter $\lambda$
>
> For the LIHC dataset, we used $\lambda = 0.45$, selected via the extended pseudo-Bayesian Information Criterion (epBIC) as in Section 6 of [19], which employed the first-order ACCORD-FBS algorithm. To assess robustness across sparsity levels, we also tested $\lambda = 0.2$ (less sparse) and $\lambda = 0.6$ (more sparse). In both cases, our method converged stably and efficiently, demonstrating reliable performance. We will add how the value $\lambda=0.45$ was chosen in the revision.
>
> (2) Line search step size
>
> Our algorithm uses a backtracking line search satisfying a sufficient descent condition. In practice, convergence is stable over a wide range of initial step sizes. For visualization purposes (e.g., Figure 3), we selected the step size schedule ($\rho_k=0.7-0.003k$ and $\sigma=0.001$) that best illustrates the rapid local convergence of the B-semismooth Newton method. This choice is purely for presentation and does not affect correctness or convergence guarantees.
>
> (3) Switching condition
>
> First of all, note that the switching is between the proximal gradient step and the B-semismooth Newton step, not between the local semismooth Netwon step and the B-semismooth Newton step. The former is absorbed into the latter; see L199.
>
> Let $|\cdot|$ be the size of a set. As noted in the paper, the Newton step becomes applicable when the submatrix $X_{\bar{\bar{\mathcal{A}}}\_k}$ is of full column rank, i.e., $\text{rank}(X\_{\bar{\bar{\mathcal{A}}}\_k}) = |\bar{\bar{\mathcal{A}}}\_k|$ . Since $\text{rank}(X_{\bar{\bar{\mathcal{A}}}_k})\leq\min(n,|\bar{\bar{\mathcal{A}}}_k|)$ always holds, a sufficient condition for full column rank is $|\bar{\bar{\mathcal{A}}}_k|\leq n$.
>
> Motivated by this, we adopt a simplified switching criterion: we switch to the Newton step when $|\bar{\bar{\mathcal{A}}}_k|\leq n$. Despite its simplicity, this rule has consistently performed well in our experiments without causing convergence issues. While more refined checks (e.g., based on condition numbers) may offer stronger theoretical guarantees, we find our criterion strikes a good balance between computational simplicity and practical robustness. We will clarify this in the revision.
>
> ### Problem 2.
>
> To assess the individual contribution of the proximal step and LCP solving modules, we re-ran the experiment of Section 4.1 for 50 iterations under three ablated configurations: disabling (i) only the proximal step, (ii) only the LCP solving step, and (iii) both. Recall that with both modules enabled, Algorithm 1 terminated within 30 iterations. The result is shown in the following table.
>
> Table : Convergence within 50 iterations under ablation of proximal and LCP updates
> |λ|Full method|No Proximal|No LCP|No Proximal + No LCP|
> |---|---|---|---|---|
> |0.1|✔  (30)|✘|✔   (30)|✘|
> |0.15|✔  (22)|✔  (26)|✔  (22)|✔  (26)|
> |0.2|✔  (14)|✔  (14)|✔  (14)|✔  (14)|
>
> At $\lambda = 0.10$, the algorithm failed to converge within 50 iterations whenever the proximal step was disabled, regardless of solving LCP. In contrast, convergence was successfully achieved in settings where the proximal step remained enabled even without LCP solving. In the original run at this $\lambda$ value, 229 out of 1000 rows were updated via the proximal step during the first iteration alone, indicating substantial reliance on this mechanism.
>
> At $\lambda = 0.15$ and $\lambda = 0.20$, the algorithm converged under all ablation settings. In these cases, at most one proximal update out of 1000 occurred in the first iteration of the original run, followed on by Newton updates thereafter. This suggests that in these sparse settings, the proximal steps may not be necessary. The catch here is, however, that we do not know the sparsity of the solution a priori, and the proximal module is needed as a safeguard.
>
> Importantly, in all configurations where convergence was achieved, the number of iterations to convergence remained comparable to that of the original results.
>
> On the other hand, the LCP step kicks in only when $|\omega_i+d_i|=\lambda$, and it is numerically subtle to declare equality. The above table suggests that Algorithm 1 is largely insensitive to this equality test. The following table summarizes the occurrences of the LCP solving step in the original run, which provides a partial reason:
> |λ|Avg. fraction of LCP per iteration|Max. fraction of LCP per Iteration|Last iteration LCP occurred|Total # iterations|
> |---|---|---|---|---|
> |0.2|0.00005|0.0011|62|159|
> |0.45|0.00004|0.0012|29|69|
> |0.6|0.00012|0.0045|31|63|
>
> This result demonstrates that the LCP solving step arises only in a negligible fraction of rows. For example, even when the Newton step is applied to as many as 10,000 rows per iteration, the average number of rows updated by LCP solving remains around 1 or fewer, as reflected by the low average/max fractions. Thus its contribution to the search direction is contained.
>
> Despite how rarely LCP occurs, it remains theoretically indispensable to ensure correctness and robustness of the algorithm before we run it on unseen data.
>
> Summary:
> - The proximal step is essential for achieving convergence.
> - Solving the LCP plays a secondary role.
> - The empirical convergence rate is largely unaffected.
>
> We will incorporate these findings into the revised version to clarify the specific roles and necessity of each module.
>
> ### Problem 3.
>
> The success of (Quasi-)Newton methods such as L-BFGS heavily depends on the assumption that the objective function is twice continuously differentiable everywhere. The ACCORD problem involves non-differentiable points due the $\ell_1$ penalty; hence, methods that do not account this non-differentiability may not be suitable. This is why we are employing semismooth Newton methods.
>
> There are a few semismooth quasi-Newton algorithms, but they are highly problem-specific (Mannel and Rund, 2021; Muoi et al., 2013). It would be great if we could develop a quasi-Semismooth Newton algorithm for ACCORD, but we believe that requires nontrivial efforts and is a subject of a separate research paper.
>
> REFERENCES
>
> Mannel, F., & Rund, A. (2021). A hybrid semismooth quasi-Newton method for nonsmooth optimal control with PDEs. *Optim. Eng.*, *22*(4), 2087-2125.
>
> Muoi, P. Q., Hào, D. N., Maass, P., & Pidcock, M. (2013). Semismooth Newton and quasi-Newton methods in weighted ℓ1-regularization. *J. Inverse Ill-Posed Probl.*, *21*(5), 665-693.
>
> ### Problem 4.
>
> The focus of the present manuscript is a semismooth Newton method for solving the ACCORD problem [19] and its globalization. Optimization of *general* convex/nonconvex functions is not the scope. We also believe an effective optimization algorithm must exploit the structure of the problem. Otherwise we may have had to stop at Theorem 2.4, leaving much room for practical issues.  As such, row-wise separability contributes to the parallelism in the algorithm and facilitates scalability without compromising theoretical rigor. Note that the semismooth Newton method remains applicable even in the absence of row-wise separability, as long as the optimality map $F$ defined by the KKT conditions is (strongly) semismooth. For example, if the $\ell_1$ penalty is replaced by the Frobenius norm, then the problem stops being separable but remains semismooth.
>
> ## Answer to Minor Problem
>
> Due to space constraints in the original submission, we could not include a detailed derivation of certain technical components, which may have hindered understanding of the algorithm’s mechanics.
>
> Here we provide a more detailed and intuitive explanation of how the generalized Jacobian is constructed, aiming to clarify the mathematical structure and logic, especially for readers less familiar with semismooth Newton methods in nonsmooth optimization.
>
> We want to solve $F(\omega^\*,d^*)=0$ where  $F:\mathbb{R}^{2p}\to\mathbb{R}^{2p}$ is defined as a nonlinear operator
> $
> F(\omega,d)=\begin{pmatrix}\mathbf{S}\omega+d\\\\\omega_1-\text{prox}\_{-\log (\cdot)}(\omega\_1+d\_1)\\\\\omega_{-1}-T_{\lambda}(\omega_{-1}+d_{-1})\end{pmatrix},\quad\text{prox}\_{-\log (\cdot)}(x) = \frac{x + \sqrt{x^2+4}}{2}.
> $
>
> Note that if $F$ is real-valued and convex, then the generalized Jacobian $\partial F$ reduces to the usual convex subdifferential.
>
> Let $F_1(\omega,d)=\mathbf{S}\omega+d$ and $F_2(\omega,d)=\begin{pmatrix}\omega_1-\text{prox}\_{-\log (\cdot)}(\omega_1+d_1)\\\\\omega_{-1}-T_{\lambda}(\omega_{-1}+d_{-1})\end{pmatrix}$.
>
> Then, it is straightforward to see that $\frac{\partial F_1}{\partial\omega}=\mathbf{S}$, $\frac{\partial F_1}{\partial d} = \mathbf{I}$ and $\frac{\partial\text{prox}\_{-\log (\cdot)}(\omega_1+d_1)}{\partial\omega_1} = \frac{\partial\text{prox}_{-\log (\cdot)}(\omega_1+d_1)}{\partial d_1} =\frac{1}{2} + \frac{\omega_1+d_1}{2\sqrt{(\omega_1+d_1)^2+4}}$.
>
> Note that for $i=2,\cdots,p$,
> $
> \frac{\partial T_\lambda(\omega_{i}+d_{i})}{\partial\omega_{i}}=\frac{\partial T_\lambda(\omega_{i}+d_{i})}{\partial d_{i}}=\begin{cases}0&|\omega_i+d_i|>\lambda\\\\ [0,1]&|\omega_i + d_i|=\lambda\\\\ 1&|\omega_i + d_i|<\lambda\end{cases}.
> $
>
> If we set $\frac{\partial T_\lambda(\omega_{i}+d_{i})}{\partial\omega_{i}}=0$ when $|\omega_i+d_i|=\lambda$, then $\frac{\partial F_2}{\partial\omega}=\mathbf{I}-\mathbf{J}$ where $\mathbf{J}=\text{diag}(J_{ii})$ is a diagonal matrix with $J_{11}=\frac{1}{2}+\frac{\omega_1+d_1}{2\sqrt{(\omega_1+d_1)^2+4}}$ and  $J_{ii}=1$ if $|\omega_i+d_i|>\lambda$, $J_{ii}=0$ if $|\omega_i+d_i|\leq\lambda$ for $i=2,\dots,p$.
>
> Thus,
> $
> \partial F(\omega,d)=\begin{pmatrix}\frac{\partial F_1}{\partial\omega}&\frac{\partial F_1}{\partial d}\\\\\frac{\partial F_2}{\partial \omega}&\frac{\partial F_2}{\partial d}\end{pmatrix}=\begin{pmatrix}\mathbf{S}&\mathbf{I}\\\\\mathbf{I}-\mathbf{J}&-\mathbf{J}\end{pmatrix},
> $
> which is equivalent to the example of the generalize Jacobian (10) in the paper.

---

### Official Review · Reviewer_fHCF · 2025-07-02

**Clarity:** 3
**Significance:** 3
**Originality:** 3
**Rating:** 4
**Confidence:** 4

**Summary:**

This paper studies partial correlation network estimation by applying graph lasso within the recently proposed ACCORD framework (Lee et al., 2019). While the method in Lee et al. (2019) is computationally scalable, it faces a bottleneck for graph lasso due to sparse-dense matrix-matrix multiplication. The authors further explore the inherent scalability of this framework by employing semismooth Newton methods and propose a scalable second-order algorithm that achieves superlinear convergence. Theoretically, they demonstrate local quadratic convergence. Simulations and real data applications comparing their method with the proximal gradient method in Lee et al. (2019) are also provided.

**Questions:**

1.  While the proposed method and algorithm does significantly improve the cur- rent algorithms based on glasso, it is known that glasso has its own problem in partial correlation network estimation. For instance, it relies on assumption of equal regularization for all nodes but this is not appropriate for real network, and the performance is not good as it may retrieves very small false edges and sensitive to outliers. See some references here:

[1] Huang. S, Jin. J, & Yao.Z. (2016). Partial correlation screening for estimating large precision matrices, with applications to classification.” Ann. Statist. 44 (5) 2018 - 2057.

[2] Cheng, C., Ke, Y., & Zhang, W. (2025). Large Precision Matrix Estimation with Unknown Group Structure. Journal of the American Statistical Association, 1–24.

Given this, how can you justify such improvement really contribute to the study of partial correlation network estimation? People are more likely to use other methods that has a fair computational cost but more accurate performance.

2. The locally quadratic convergence result is quite attractive. How can we heuristic under stand it? What is the intuition behind this quadratic convergence?

**Ethical Concerns:**

["NO or VERY MINOR ethics concerns only"]

**Final Justification:**

I would like to thank the authors for prepare a carefully-written rebuttal and a carefully-written response to my most recent questions.
I wish to clarify that, by accuracy, I meant numerical experiments. The authors pointed to some papers on GLASSO on the literature,
but those papers focus on theory. My experience is that, numerically, GLASSO is quite poor, compared to other methods. Overall, I think the paper focus on computation side, but does not adequate address the accuracy the procedure. I would encourage an extensive numerical study, where the authors can compare with representative methods in the literature. I will keep my rating.

**Limitations:**

Yes

**Quality:**

3

**Strengths And Weaknesses:**

Strength: The paper is well-organized with some theory to support their method and demonstrate the phenomenon appears in simulation. The improvement compared to Lee et al., 2019 is significant.

Weakness:

1. The writing of theoretical part is hard to understand. More explanations on the lemma and theorem and clearly state the connection of the lemmas could help improve the readability given the abundant notations crowded in lemmas or theorems.

2.  The numerical part only compare their method with the one in Lee et al., 2019. How about other algorithms and also the algorithms beyond glasso framework? In addition, only convergence rate or computational cost is compared, nothing is
reported about accuracy.

3. The motivation of the paper is this: Glasso is a great method, therefore, we wish to improve it. However, despite the popularity of the glasso, study on the glasso is largely theoretical, with limited numerical success for high dimensional data (simulated and real). In fact, it was reported that the glasso estimation of the precision matrix tends to be overly sparse, and alternative approaches may have much better performance.    My question is,  if the glasso is not an effective algorithm, how to justify the work that tries to improve its (and only its) computing side? It seems that the more important problem is to improve its accuracy.  In fact, the paper only investigate the computing speed of the proposed method, without investigation on the estimation accuracy.

---

> ### Author Rebuttal · Authors · 2025-07-31
>
> ### Answer to W1
>
> We appreciate your comments on the clarity of our theoretical development. We acknowledge that the presentation of Lemma 3.3, Theorem 3.4, and Theorem 3.8 may have been hard to follow due to dense notation and limited space. We welcome the opportunity to clarify the logical structure and agree that more detailed explanations would improve readability.
>
> The theoretical foundation of our method is presented in two parts:
> - In Section 2, we study the local convergence of the semismooth Newton method under the standard framework based on generalized Jacobians. In particular, Theorem 2.4 shows that if a generalized Jacobian is nonsingular near a root of a (strongly) semismooth function, then the Newton iteration converges locally quadratically. To ensure this condition in our problem, Proposition 2.5 provides a sufficient condition based on the rank of a submatrix of the design matrix, which is linked to the model identifiability.
> - Section 3 extends the analysis to ensure global convergence of the semismooth Newton-type method when applied to the  ACCORD objective. While the local theory guarantees fast convergence near the solution, it does not inform us how to choose a suitable initial point. To address this, we propose a damping strategy and work with the Bouligand (B-) derivative, a specific choice of the generalized Jacobian. This leads to the development of the B-semismooth Newton method, which is amenable to globalization.
>
> We summarize below how the key lemmas and theorems connect within this framework:
>
> - Proposition 2.5 provides a sufficient condition for the nonsingularity of generalized Jacobians, thereby ensuring local quadratic convergence per Theorem 2.4.
> - Lemma 3.3 reformulates the B-semismooth Newton direction as a solution to a linear complementarity problem (LCP). This result is key to formulating the Newton step as a tractable numerical subproblem: solving the B-derivative equation is only a descriptive step in general B-semismooth Newton algorithm. Here we provide a constructive way to solve it - solve a well-defined LCP in reduced dimension.
> - Theorem 3.4 shows that the well-defined LCP admits a unique solution, so the B-semismooth Newton iteration is indeed well-defined.
> - Corollary 3.5 confirms local quadratic convergence of the B-semismooth Newton iteration under the same rank condition as Proposition 2.5.
> - Theorem 3.8 establishes global convergence of Algorithm 1. Even when the Newton direction is undefined (e.g., due to local rank deficiency), the algorithm uses a proximal gradient algorithm, which occurs only finitely many times. Eventually, the method enters the region where local quadratic convergence (guaranteed by Theorem 3.4 and Corollary 3.5) takes over.
>
> In the revised version, we will add brief guiding remarks before theorems and propositions to explain their roles and improve the readability of the theoretical development.
>
> ### Answer to W2
>
> We would like to clarify that the scope of the present manuscript is enhancing the algorithm for solving the ACCORD optimization problem (3) proposed in [19]. ACCORD is a pseudolikelihood method designed for scalable estimation of partial correlation networks, distinct from the the graphical lasso (glasso) framework in learning partial correlation graphs, and their comparison with respect to convergence property and model performance is already presented in [19] along with other preceding pseudolikelihood-based models. Regarding the improvement of the solution algorithm for ACCORD, however, no alternative beyond the proximal gradient method in [19] has been proposed, to the best of our knowledge. As such, we  focus on comparing the computational performance between this baseline algorithm and our semi-smooth Newton approach. If there is no time and memory limits, then both algorithms produce the same optimal solution for the same data, which makes the present comparison meaningful.
>
> ### Answer to W3
>
> As we pointed out above, glasso and ACCORD are different frameworks/methods for learning partial correlation graphs from data. As we mentioned in Introduction, ACCORD is closer to pseudolikelihood-based methods such as SPACE [27] or CONCORD [17] than glasso, which are known to be more robust to non-normality. Also, It is *not* that the glasso estimates are overly sparse and the statistical performance of glasso is bad, but that it is not *computationally* tractable in modern, ultra-high dimensional applications, say when $p > 100,000$. ACCORD [19] is a recently proposed, viable alternative to glasso, yielding a  tractable and scalable optimization problem in computing the estimate with a similar statistical performance to glasso. While the statistical properties are studied in depth in [19], in our view, the optimization algorithm in [19], which is a first-order proximal gradient algorirhtm, is not sufficiently efficient. So the present manuscript focuses on the computational aspect of the ACCORD framework, and exploits the semismooth nature of the optimization problem to produce faster and more scalable algorithm. We demonstrate that our algorithm reaches the same solution as [19] within substantially fewer iterations and lower runtime—an important practical advantage for large-scale partial correlation graph estimation. For the statistical performance of ACCORD estimator, including the accuracy (estimation error) measured in various norms, please refer to [19].
>
> ### Answer to Q1
>
> Again, we emphasize that our contribution lies in improving the computational performance of the estimation algorithm, rather than in the methodology itself. Nonetheless, the “equal regularization for all nodes” problem can be easily addressed by introducing an individual regularization coefficient (hyperparameter) $\lambda_{ij}$ for each *edge*. The theory of [19] applies with essentially no modification as long as $\lambda_{ij}$’s diminish with $n$ and $p$ at the same rate. Refinements like SLOPE (Bogdan et al., 2015) to control false discovery rates can also be made easily.
>
> The method of Cheng, Ke, and Zhang (2025) is essentially the symmetric lasso loss (Friedman et al., 2010) with sparse-group lasso penalty (Simon et al, 2013), and the focus of the paper is how to find groups from data. As pointed out in [17], symmetric lasso is a nonconvex loss function and its convergence is problematic. In contrast, the ACCORD loss is convex and the sparse-group lasso penalty can be easily imposed to make the overall optimization problem convex. In many multi-omics studies, groups are known a priori in biological basis. Statistical analysis of the resulting estimator, however, requires a distinct set of assumptions, which is common in the sparse learning literature (see Wainwright, 2019).
>
> The method of Huang et al. (2016) requires frequent access to *all* the principal submatrices of the sample covariance matrix. When $p > 100,000$ as we focus, constructing the $p\times p$ sample covariance matrix itself is troublesome, which often requires a distributed storage [19]. In this setting, access to principal submatrices faces a communication bottleneck. Our algorithm only requires storing the $n \times p$ data matrix $X$ and $n \ll p$. Further, each row of $X$ is independently fed to the algorithm to run in parallel. Methodologically, the method of Huang et al. (2016) depends on the ordering of the variables. Different ordering may result in quite different outcomes.
>
> **REFERENCES**
>
> Bogdan, M., Van Den Berg, E., Sabatti, C., Su, W., & Candès, E. J. (2015). SLOPE—adaptive variable selection via convex optimization. *Ann. Appl. Stat.* 9 (3), 1103.
>
> Friedman, J., Hastie, T. and Tibshirani, R. (2010) *Applications of the lasso and grouped lasso to the estimation of sparse graphical models*. Technical Report. Stanford University, Stanford.
>
> Simon, N., Friedman, J., Hastie, T. and Tibshirani, R., 2013. A sparse-group lasso. *J.  Comput. Graph. Stat.*, *22*(2), pp.231-245.
>
> Wainwright, M. J. (2019). *High-dimensional statistics: A non-asymptotic viewpoint* (Vol. 48). Cambridge University Press.
>
> ### Answer to Q2
>
> Thank you for your insightful question. The locally quadratic convergence behavior is indeed one of the key advantages of our proposed second-order method.
>
> To build intuition, let us contrast this with first-order methods such as gradient descent. In first-order algorithms, each iteration at best reduces the error by a constant multiplicative factor, i.e., convergence is sublinear and becomes linear when the objective is strongly convex. Practically, this means that each iteration improves the solution by approximately one significant digit. As a result, achieving high-precision solutions requires a large number of iterations, especially near the optimum.
>
> In contrast, second-order methods such as the semismooth Newton method exploit curvature information via (generalized) Hessians or generalized Jacobians. Once sufficiently close to the solution, these methods enjoy quadratic convergence, meaning that the number of accurate digits in the solution approximately doubles for each iteration.
>
> This dramatic acceleration near the solution is particularly beneficial for solving non-smooth problems like ours, where naive first-order approaches often suffer from slow convergence or stagnation. The value of the first-order method is that it is global — it converges (after many iterations) to the solution wherever it begins, while for second-order methods to quadratically converge, they should start near the solution. So we combined the two in order to globalize the semismooth Newton algorithm so that it converges from everywhere. In our experiments, this theoretical advantage translates into practice: the algorithm consistently converges to high-precision solutions within a few tens of iterations, even for high-dimensional instances.
>
> We hope this clarifies the intuition behind the observed quadratic convergence.

---

> > ### Comment · Reviewer_fHCF · 2025-08-08
> > **Response to Author**
> >
> > Thanks for your response to my raised points. It does help clarify some of my points, but I think you missed some of the important points.
> >
> > You seem to say that your focus is on computation. However, this is only one (and arguably important) point, which I have no
> > problem with it. However, the more important point is that, your estimator needs to be accurate.
> >
> > As I pointed out in my previous arguments, the glasso is shown to be very inaccurate in high dimensional setting. You clarified that your method is different from glasso. That is fine. But you need to show your method is accurate.
> >
> > In particular, it was shown that for many real data, the glasso estimate for the precision matrix is way too more sparse than expected. For example, many rows and columns (excluding the single entry on the main diagonal of the matrix) are estimated as 0. This is a major concern.
> >
> > My point is, it is great to be able to compute an algorithm, but it is even more important to check your algorithm is accurate. Unfortunately, I was not convinced at the latter.
> >
> > Would you be so kind to clarify this point? Thanks.

---

> ### Author Response · Authors · 2025-08-09
> **Re : Response to Author**
>
> If you mean by “accuracy” the statistical consistency of an estimator, such as that the error measured by the Frobenius norm, spectral norm, or $\ell_1$ norm between the difference between the estimated precision matrix and the true, unknown precision matrix, there are well-established results in the GLASSO literature; see Ch. 11 in Wainwright (2019) or Ravikumar et al. (2011). These results in general state that if the data distribution is reasonable (e.g., sub-Gaussian) and the true precision matrix is sparse in a certain sense, then, with the full Gaussian log-likelihood as the loss function and an $\ell_1$-type penalty that encodes the assumed sense of sparsity, an appropriately chosen penalty coefficient $\lambda$ yields a consistent estimator; i.e., the difference of the desired aspect between the estimated and true precision matrices tends to zero as both the sample size $n$ and the dimension $p$ tends to infinity in a certain manner (usually $\sqrt{\log p/n} \to 0$). The results vary in what aspect you would like to see.  For ACCORD, such results are presented in [19, Sect. 4] for the “element sparsity model” where the plain $\ell_1$ norm is appropriate as a penalty, backed with the simulation studies [19, Sect. 5]. This model is what we (implicitly) deal with in the present manuscript. In short, ACCORD is on par with GLASSO in terms of statistical performance, as established in [19]. Please see also our reply to Reviewer r5ah (Re: Additional major concern).
>
> Every statistical method has an assumed model of reality. If the data-generating process (DGP) fits the model assumptions to a certain degree, then the method will produce an “accurate” estimate. If DGP does not fit, then it will fail. No method is universal. The method of Cheng, Ke, and Zhang (2025) that you raised will work if the dataset has an underlying “group sparsity” structure. But if it is only “elementwise sparse,” then GLASSO or ACCORD is likely to work better. Likewise, the method of Huang, Jin, and Yao (2016) is likely to work better than GLASSO or ACCORD if the assumed ordering of the variables is correct, but isn’t even if the “element sparsity” holds.
>
> Perhaps what you meant by “the glasso estimate for the precision matrix is way too more sparse than expected. For example, many rows and columns (excluding the single entry on the main diagonal of the matrix) are estimated as 0“ is the case when the dimension is high and for small values of the penalty coefficient $\lambda$ the optimization algorithm fails but it manages to compute the estimate for large values of $\lambda$, resulting in nearly diagonal estimates. This is precisely the phenomenon reported in [19]: “BigQUIC was not able to complete the computation of the precision matrix for a wide range of the regularization parameter $\lambda$, except for those that yielded diagonal matrices [19. p.16].” As pointed out in L40—L47 of the manuscript, this phenomenon is due to the inherent computational bottleneck in the GLASSO optimization problem. ACCORD mitigates this bottleneck by adopting a novel, computation-friendly loss function while maintaining statistical performance. Our work significantly improves the convergence speed and makes ACCORD more accessible to lay users. See also our Re: Additional major concern to Reviewer r5ah.
>
> In summary, ACCORD estimator is “accurate” in the situation that it is meant to be used, and your negative experience with GLASSO is likely due to its inherent computational bottleneck, which ACCORD methodology does not suffer from.
>
> **REFERENCES**
>
> Wainwright, M. J. (2019). *High-dimensional statistics: A non-asymptotic viewpoint* (Vol. 48). Cambridge University Press.
>
> P. Ravikumar, M. J. Wainwright, G. Raskutti, and B. Yu (2011). High-dimensional covari-
> ance estimation by minimizing ℓ1-penalized log-determinant divergence. *Electron. J.
> Stat.*, 5:935 – 980.

---

### Official Review · Reviewer_r5ah · 2025-07-03

**Clarity:** 3
**Significance:** 3
**Originality:** 3
**Rating:** 3
**Confidence:** 2

**Summary:**

This paper proposes a new semismooth Newton method for efficiently estimating partial correlation networks using an ℓ₁-regularized pseudolikelihood-based framework. Building on the existing ACCORD framework (which itself is scalable but relies on slower first-order methods), the authors derive a second-order optimization algorithm that significantly improves convergence speed and reduces reliance on expensive hardware like supercomputers.
The method is applied to both synthetic and real-world datasets, showing improved performance over the state-of-the-art proximal gradient method (ACCORD-FBS) in terms of convergence rate, iteration count, and wall-clock time.

**Questions:**

1.  How sensitive is the method to the choice of regularization parameter λ, particularly in real-world datasets?
2. Have you tested the algorithm on non-Gaussian data distributions? How does performance compare in those settings? Include experiments on non-Gaussian synthetic data (e.g., t-distributions or contaminated normals) to validate robustness.

**Ethical Concerns:**

["NO or VERY MINOR ethics concerns only"]

**Final Justification:**

I cannot access the contribution of an efficient second-order algorithm for ACCORD, given that ACCORD [19] has not shown its wide applicability and correctness yet, which is arXiv paper beyond the reviewing scope of this paper.

To clarify, I do acknowledge the technical difficulty of developing a second-order solver that is more efficient and with provable performance for ACCORD, that's why I initially gave 4. But I find that difficulty does not necessarily mean a significant contribution, given that the significance of the original algorithm ACCORD has not been shown yet, which is what I did not realize before the rebuttal. That's why I lowered my score to 3. However, my confidence score remains low as before.

**Limitations:**

yes

**Quality:**

3

**Strengths And Weaknesses:**

Strengths:

1. The authors develop a semismooth Newton approach with a strong theoretical foundation.
2. The algorithm is shown to have locally quadratic convergence, which is a major improvement over linear convergence of previous methods.
3. The paper provides clear derivations, conditions for global convergence, and strategies for damping and line search.

Weekness

1. The method assumes submatrices like X_A have full column rank. While reasonable in many high-dimensional biological datasets, this may not hold in low-sample or noisy regimes. Include a robustness analysis or empirical study on how violations of this assumption affect convergence and solution accuracy would be helpful
2. One motivation for pseudolikelihood methods is robustness to non-Gaussian data, but this is not empirically evaluated in the paper.

---

> ### Author Rebuttal · Authors · 2025-07-31
>
> ### Answer to Weakness 1
>
> In fact the desired robustness study has been already presented in Section 4.2. We conducted systematic experiments by varying both the graph density (ratio of the number of edges in the true graph to that of the complete graph) and the regularization parameter $\lambda$; see Tables 1 and 2. These experiments allow us to examine how the size of the active set—and by implication, the likelihood of rank-deficiency in $X_{\bar{\bar{\mathcal{A}}}_k}$— which in turn affects the applicability of the semismooth Newton step and overall convergence.
>
> The results reveal a clear pattern:
>
> - Our algorithm converges reliably within 300 iterations in the majority of settings;
> - Failure occurs when the true graph is dense (e.g., density of 0.10) and the regularization parameter $\lambda$ is too small to sufficiently regularize the solution. This leads to a large active set, increasing the chance of rank-deficiency of $X_{\bar{\bar{\mathcal{A}}}_k}$  and preventing semismooth Newton steps from being applied.
>
> To further assess robustness under more complex structural variations, Appendix F presents additional experiments on hub networks and scale-free graphs, which exhibit diverse local connectivity patterns (Section 4.2 is about uniformly random graphs). Importantly, we observe that convergence behavior remains consistent across these graph types when the overall graph density is held fixed. This strongly suggests that the convergence of our method—and the applicability of the semismooth Newton step—is primarily governed by global sparsity (active set size) rather than local graph topology.
>
> Taken together, these findings provide empirical supports that our globalized algorithm remains robust in realistic settings as long as the model is appropriately regularized.
>
> Obtaining a statistical guarantee of continuation strategies *a la* [7] under noisy regime is challenging even in the regression setting [15]. Failure to converge within, say, 300 iterations, and/or dominance of the proximal gradient steps in our globalized algorithm as examined in Section 4.2 and Appendix F may suggest a stopping criterion that adapts to the sparsity level of noisy data, while further study is warranted. Thank you for the insightful comment!
>
> ### Answer to Question 1
>
> Per your suggestion, we have investigated the convergence behavior across a range of regularization parameters $\lambda$ under the same experimental settings described in the paper, focusing on real-world datasets.
>
> Specifically, for the LIHC dataset, we used the value $\lambda = 0.450$, selected via the extended Bayesian Information Criterion proposed in [19]. To further assess the sensitivity of our algorithm with respect to the choice of $\lambda$, we systematically varied $\lambda$ from 0.325 to 0.575 with increment of 0.025.
>
> The table below presents the number of iterations, the number of nonzero elements in the estimated precision matrix, and its corresponding density.
>
> | λ | Iterations | Nonzeros | Density (%) |
> | --- | --- | --- | --- |
> | 0.325 | 103 | 5,494,644 | 5.89 × 10⁻³ |
> | 0.350 | 69 | 4,806,952 | 5.15 × 10⁻³ |
> | 0.375 | 78 | 4,214,966 | 4.52 × 10⁻³ |
> | 0.400 | 67 | 3,699,259 | 3.96 × 10⁻³ |
> | 0.425 | 94 | 3,248,215 | 3.48 × 10⁻³ |
> | 0.450 | 64 | 2,852,968 | 3.06 × 10⁻³ |
> | 0.475 | 67 | 2,505,706 | 2.68 × 10⁻³ |
> | 0.500 | 71 | 2,200,810 | 2.36 × 10⁻³ |
> | 0.525 | 56 | 1,930,194 | 2.07 × 10⁻³ |
> | 0.550 | 66 | 1,692,240 | 1.81 × 10⁻³ |
> | 0.575 | 65 | 1,481,495 | 1.59 × 10⁻³ |
>
> Based on the leading significant digits of the estimated sparsity levels (density), the variation remains modest within the range $\lambda \in [0.4, 0.5]$, suggesting that this interval corresponds to a region where the final sparsity pattern is relatively insensitive to the choice of the “optimal” $\lambda$.
>
> We also observed that our algorithm consistently converges within a few tens of iterations across all tested values of $\lambda$. Although the number of iterations tends to slightly increase as $\lambda$ decreases, the overall convergence behavior remains stable.
>
> ### Answer to Weakness 2 and Question 2
>
> We investigated the performance of ACCORD on simulated data with $n=500$ and $p=1{,}000$, generated from a multivariate $t$-distribution with degrees of freedom $\nu = 5$, where the graph structure of the precision matrix $\Theta^*$ was defined by an Erdős–Rényi graph with 1,000 randomly selected edges. We present the overall result from 50 replications following [19], including the mean (standard deviation) of the Area Under Precision-Recall Curve (AUPRC) obtained from multiple values of $\lambda$ and number of True positive (TP) / False Positive (FP) edges detected for the selected $\lambda$. The $\lambda$ was selected with the extended psuedo-Bayesian Information Criterion (ep-BIC) for ACCORD and the plain BIC for GLASSO, following [19]. The result is as follows.
>
> | Method | AUPRC | # TP edges | # FP edges |
> | --- | --- | --- | --- |
> | ACCORD | 0.722 (0.082) | 704 (28.9) | 264 (277.9) |
> | GLASSO | 0.695 (0.098) | 698 (26.6) | 399 (712.9) |
>
> We could observe that by comparing AUPRC, ACCORD showed better overall performance on detecting the precision graph. Comparing the graphs selected with (ep)-BIC, ACCORD detected significantly less false positive edges than GLASSO while both method detected similar number of true positive edges.
>
> We note that the statistical performance of the ACCORD model is not our primary focus of the present manuscript, as the aim of the study is to develop a second-order algorithm that solves the ACCORD optimization problem (3) order of magnitude faster than the proximal gradient method proposed in [19].

---

> > ### Comment · Reviewer_r5ah · 2025-08-06
> > **Additional major concern**
> >
> > I appreciate the response from the authors, and the Q1 and W1 have been addressed.
> >
> > However, the authors' response regarding "statistical performance of the ACCORD model is not our primary focus of the present manuscript, as the aim of the study is to develop a second-order algorithm that solves the ACCORD optimization problem (3) order of magnitude faster'' raises **additional major concerns** that I had not noticed in the initial review
> >
> > As the authors mentioned, "the novelty of the present paper is to develop a second-order method to solve for ACCORD", it is my understanding that such a contribution is only significant if ACCORD is an algorithm/model that is widely used and shows effectiveness for various network estimation problems ( like GLASSO). However, I also went ahead and read [19], which is an arXiv submission, and after reading [19], I doubt the contribution of ACCORD, whose performance seems to be only tested on synthetic and small-scale datasets. Hence, while I acknowledge the effort and the length of the mathematical derivation, I have a critical question what the motivation is to develop an efficient solver for ACCORD if the performance of ACCORD has not been shown in wide application? I would appreciate the authors' further clarification on this.

---

> > > ### Author Response · Authors · 2025-08-08
> > > **Re: Additional major concern**
> > >
> > > In our view, the major contribution of the ACCORD methodology [19] is its feasibility of analyzing truly large-scale data demanded in clinical multi-omics in a statistically valid fashion. Section 6 in [19] is devoted to a thorough network analysis of the liver cancer data of dimension $p=285,358$. The key research question there is to identify the repressive effect of epigenetic elements in the co-expression patterns of RNAs. This requires inclusion of the DNA methylations data into the RNA expression data. Since the former alone has dimension 269,396, without a computational method that scales up to this level, the desired analysis is impossible. However, [19] reports “BigQUIC was not able to complete the computation of the precision matrix for a wide range of the regularization parameter $\lambda$, except for those that yielded diagonal matrices [19. p.16].” Recall that BigQUIC is one of the most scalable implementations of GLASSO. So GLASSO canNOT be used to analyze this kind of omics data of clinical significance. The statistical performance of ACCORD in on par with GLASSO, either in theory ([19, Sect. 4]) or in simulation studies ([19, Sect. 5]). At the scale of the liver cancer dataset, ACCORD is the only runner on the track, so comparison with GLASSO is moot. And for clinical/biological datasets like this, clinical significance should be the primary measure of performance. The analysis of [19, Sect. 6] shows this (”In summary, the evaluation shows that the network derived from the dual-omic data generates more robust relational hypotheses than the network derived from the transcriptomic data alone. By directly estimating conditional dependence structure from ultrahigh-dimensional data, spurious associations can be screened out and the filtered data improves the quality of biological inference of regulatory relationships. It goes without saying that the superior performance in the present case study comes from the scalable computation enabled by the HP-ACCORD framework [19, p.18].”)
> > >
> > > Nevertheless, the scaliability in [19] is achieved at the cost of implementing the MPI version of the ACCORD-FBS algorithm (using a special linear algebra library) on Cray CS500/XC40 supercomputer with many highly performing computational nodes, and running it for many hundreds of iterations. Many researchers cannot afford such computing and human resources to run the algorithm but the demand for such a large-scale clinical multi-omics study is still high.
> > >
> > > Here comes our contribution. We identify the rowwise separability of the  problem and exploit it to downsize the computation to the level of GPU workstations. Our own implementation of ACCORD-FBS allows us to analyze the same dataset on much cheaper and accessible systems. (Don’t forget PyTorch vs MPI!) We then go further to exploit the semismoothness of the problem to develop a semismooth Newton algorithm that converges within a few tens of iterations. That’s an order of magnitude improvement over ACCORD-FBS. Since GPU workstations are less powerful than Cray machines, the wall-clock time for the same hundreds of iterations of ACCORD-FBS is much longer. So an optimization algorithm that converges an order of magnitude faster is more impactful on GPU machines and contributes to the practicality of the methodology. We believe that our contribution will facilitate wide application of the ACCORD methodology.

---

> > > > ### Comment · Reviewer_r5ah · 2025-08-09
> > > >
> > > > Thank you for your response and clarification. However, my concern still remains as I cannot access the contribution of an efficient second-order algorithm for ACCORD, given that ACCORD [19] has not shown its wide applicability and correctness yet, which is arXiv paper beyond the reviewing scope of this paper.
> > > >
> > > > To clarify, I do acknowledge the technical difficulty of developing a second-order solver that is more efficient and with provable performance for ACCORD, but I find that difficulty does not necessarily mean a significant contribution, given that the significance of the original algorithm ACCORD has not been shown yet.

---

### Decision · Program_Chairs · 2025-09-17

**Decision:**

Accept (poster)

**Comment:**

The paper considers partial correlation network estimation. The classical approach, graphical lasso, scales cubically and hence struggles to handled datasets where p>1000. Instead, the recently proposed ACCORD framework solves an equivalent problem that is much easier to solve for large p. Prior work based on ACCORD solve the equivalent problem using proximal gradient, which enjoys scalable iterations but converge too slowly. This paper, therefore, develops a bespoke semismooth-Newton method for solving the equivalent problem. The method enjoys quadratic local convergence and can be implemented to be scalable. Experiments on synthetic and large liver-cancer genomics show large speedups from far fewer iterations compared to proximal-gradient baselines. The method also reduces hardware demands compared with prior MPI/cluster settings.

Concerns mostly focus on scope and presentation: clarifying assumptions (e.g., rank conditions), documenting hyperparameter choices and switch criteria, broadening comparisons (e.g., quasi-Newton), and adding robustness studies (non-Gaussian data). The rebuttal added sensitivity analyses, ablations, and non-Gaussian results, and clarified how ACCORD addresses glasso bottlenecks at extreme dimension.